# SELF-SUPERVISED SPATIAL REPRESENTATIONS FOR DEEP REINFORCEMENT LEARNING

## ABSTRACT

Recent reinforcement learning (RL) methods have found extracting high-level features from raw pixels with self-supervised learning to be effective in learning policies. However, these methods focus on learning *global* representations of images, and disregard *local* spatial structures present in the consecutively stacked frames. In this paper, we propose a novel approach that learns self-supervised *spatial* representations ($\mathbf{S}^3\mathbf{R}$) for effectively encoding such spatial structures in an unsupervised manner. Given the input frames, the spatial latent volumes are first generated individually using an encoder, and they are used to capture the change in terms of spatial structures, *i.e.*, flow maps among multiple frames. To be specific, the proposed method establishes flow vectors between two latent volumes via a supervision by the image reconstruction loss. This enables for providing plenty of local samples for training the encoder of deep RL. We further attempt to leverage the spatial representations in the self-predictive representations (SPR) method that predicts future representations using the action-conditioned transition model. The proposed method imposes similarity constraints on the three latent volumes; warped *query* representations by estimated flows, predicted *target* representations from the transition model, and *target* representations of future state. Experimental results on complex tasks in Atari Games and DeepMind Control Suite demonstrate that the RL methods are significantly boosted by the proposed self-supervised learning of spatial representations. The code is available at `https://sites.google.com/view/iclr2022-s3r`.

## 1 INTRODUCTION

Deep reinforcement learning (RL) has been an appealing tool for training agents to solve various tasks including complex control and video games (François-Lavet et al., 2018). While most approaches have focused on training deep RL agent under the assumption that compact state representations are readily available, this assumption does not hold in the cases where raw visual observations (*e.g.* images) are used as inputs for training the deep RL agent. Without designing an effective algorithm that uses model-based policy and value improvement operators (Schrittwieser et al., 2021), or without attempting to use additional image augmentation (Laskin et al., 2020a; Kostrikov et al., 2020), learning visual features from raw pixels only using a reward function might fail to learn good features in terms of the performance and sample efficiency.

To address this challenge, a number of deep RL approaches (Sermanet et al., 2018; Dwibedi et al., 2018; Anand et al., 2019; Laskin et al., 2020b; Mazoure et al., 2020; Stooke et al., 2020; Schwarzer

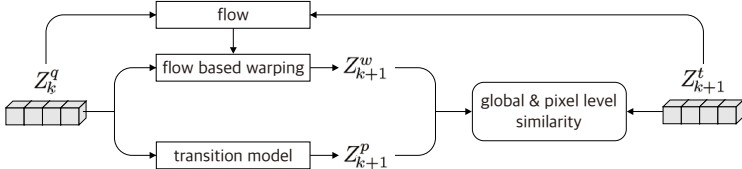

Figure 1: Abstract form of the $\mathbf{S}^3\mathbf{R}$ method: Warped representation $Z^w_{k+1}$ and predicted representation $Z^p_{k+1}$ are obtained from query encoder representation $Z^q_k$ at frame $k$ by flow based warping and transition model, respectively. Both global-level and pixel-level similarity are then imposed with target encoder representation $Z^t_{k+1}$ at future frame $k+1$.

et al., 2021) leverage the recent advance of self-supervised learning which effectively extracts high-level features from raw pixels in an unsupervised fashion. In Laskin et al. (2020b); Stooke et al. (2020), they propose to train the convolutional encoder for pairs of images using a contrastive loss (van den Oord et al., 2018). For training the RL agent, given a query and a set of keys consisting of positive and negative samples, they minimize the contrastive loss such that the query matches with the positive sample more than any of the negative samples (Laskin et al., 2020b; Stooke et al., 2020). While the parameters of the query encoder are updated through back-propagation using the contrastive loss (van den Oord et al., 2018), the parameters of the key encoder are computed with an exponential moving average (EMA) of the query encoder parameters. The output representations of the query encoder are passed to the RL algorithm for training the agent. Schwarzer et al. (2021) proposes the self-predictive representations (SPR) method that trains the RL agent by leveraging the query and positive sample only, following Grill et al. (2020) that achieves state-of-the-arts performance without the negative samples in the self-supervised learning. Especially, it extends a model-free RL agent by adding the transition model that predicts its own latent representation multiple steps into the future (Schwarzer et al., 2021). The predicted future representations and the representations for future states computed using a target encoder serve as the query and positive sample, respectively. These approaches have shown compelling performance and high sample efficiency on the complex control tasks when compared to existing image-based RL approaches (Kaiser et al., 2019; van Hasselt et al., 2019; Kielak, 2020).

While these approaches can effectively encode the *global* representations of images with the self-supervised representation learning, there has been no attention on the *local* spatial structures present in the consecutively stacked images. Our key observation is that spatial deformation, *i.e.*, the change in terms of the spatial structures across the consecutive frames, can provide plenty of local samples for training the RL agent. A two-frame flow estimation (Dosovitskiy et al., 2015; Ilg et al., 2017; Jonschkowski et al., 2020), which has been widely used for video processing and recognition in computer vision, can be an appropriate tool in modeling the spatial deformation. In this work, we propose a novel approach, termed self-supervised *spatial* representations ($S^3R$), that learns spatial representations for effectively encoding the spatial structures in a self-supervised fashion. Note that we use the term 'spatial representations' to indicate a set of feature maps extracted over frames for inferring locally-varying flow maps. The spatial representations generated from an encoder are used to predict the flow maps among the input frames by minimizing an image reconstruction loss in a self-supervised manner. A flow-based warping is then applied to generate future representations. We further extend our framework by leveraging SPR method (Schwarzer et al., 2021). As depicted in Figure 1, we impose similarity constraints on the three representations; query representations warped by the estimated flow maps, self-predicted target representations from the transition model (Schwarzer et al., 2021), and target representations of future state.

Note that Shang et al. (2021) encodes the temporal information by concatenating latent differences of input frames, but the simple substraction operation has certain limitations in effectively capturing the spatial deformation. Contrary to Amiranashvili et al. (2018) in which the flow map is directly fed into RL algorithms with a stack of images, our method leverages the flow maps to effectively encode the spatial representations of the images and warp spatial representations at the future state.

Our contribution is summarized as follows.

- Our method learns the spatial representations using the self-supervised flow model for encoding *local* spatial structures from the consecutive frames used in RL algorithms.
- We propose to impose the similarity constraint on the spatial representations as well as the global representations, providing plenty of supervision for training the encoder of deep RL.
- We compute the future representations through the flow-based warping for imposing the similarity constraint with target representations.

## 2 RELATED WORK

**Self-supervised Representation Learning**: The self-supervised representation learning aims to learn general features from large-scale unlabeled images or videos without expensive data annotations. The contrastive methods have achieved state-of-the-art performance in the self-supervised representation learning. The contrastive learning aims to bring positive samples closer while sep-

arating negative samples from each other (Hadsell et al., 2006). Wu et al. (2018) formulate the contrastive learning as a non-parametric classification problem at the instance level, and propose to learn visual features with the memory bank and noise contrastive estimation (NCE) (Gutmann & Hyvärinen, 2010; Mnih & Kavukcuoglu, 2013). The method in van den Oord et al. (2018) proposes a probabilistic contrastive loss, called InfoNCE, for inducing representations by leveraging positive and negative samples. The InfoNCE loss has widely been adopted in Chen et al. (2020); He et al. (2020); Hénaff et al. (2020); Tian et al. (2020). Chen et al. (2020) present a simple framework for contrastive self-supervised learning without specialized architecture (Bachman et al., 2019; Hénaff et al., 2020) or memory bank (Wu et al., 2018), but it requires a large batch size for using enough negative samples when computing the InfoNCE loss (van den Oord et al., 2018). He et al. (2020) propose to build a dynamic dictionary with a queue to avoid the use of large batches when collecting negative samples, and also uses the moving averaged (momentum) encoder for target data (positive and negative samples of query data). Grill et al. (2020) use the momentum encoder to produce representations of the targets as a means of stabilizing the bootstrap step. This enables for learning the representations with only positive samples, which are generated by data augmentation, for a given query without the need to carefully set up negative samples. The method in Chen & He (2020) further extends this idea by using only stop-gradient operation without using the momentum update. While these approaches focuses on learning global representations of a single image, our method proposes to learn spatial representations for effectively encoding the spatial structures (*i.e.*, flow map) in the consecutive images.

**Self-supervised Representation Learning in Deep RL**: Representation learning is crucial for RL algorithms to learn policies with high-dimensional visual observations. The future prediction conditioned on the past observations and actions serves as auxiliary tasks to improve the sample efficiency of model free RL algorithms. Gelada et al. (2019) train a transition model to predict representations of future states together with a reward prediction loss. Guo et al. (2020) present Predictions of Bootstrapped Latents (PBL) that builds on multi-step predictive representations of future observations for deep RL. The method in Schwarzer et al. (2021) propose Self-Predictive Representations (SPR) based on an action-conditioned transition model that can predict future representations computed using a target (momentum) encoder. Our method attempts to predict the future representations through a warping operation using flow maps computed from the spatial representations of the consecutive frames.

Contrastive learning has been used to extract desired latent representations of visual observations used in the RL algorithms. For training robot agents, Sermanet et al. (2018) present the time-contrastive networks (TCN) that train viewpoint-invariant representations using a metric learning. This work was extended in Dwibedi et al. (2018) by embedding multiple frames at each timestep for learning task-agnostic representations such as position and velocity attributes in continuous control tasks. In Anand et al. (2019), the representations for RL algorithms are learned by maximizing mutual information (Hjelm et al., 2019) across spatially and temporally distinct features of an encoder of visual observations. Schwarzer et al. (2021) leverage the self-supervised learning (Grill et al., 2020) for imposing the similarity constraint between self-predictive and target representations. Laskin et al. (2020b) introduce Contrastive Unsupervised representations for Reinforcement Learning (CURL) that learns the representations from visual inputs using the InfoNCE loss (van den Oord et al., 2018). Stooke et al. (2020) present Augmented Temporal Contrast (ATC) using image augmentations and InfoNCE loss (van den Oord et al., 2018) for representation learning, and decouples it from policy learning. From a different perspective, Hansen et al. (2020) propose to adapt the policy network through self-supervised representation learning in unseen environments where it is difficult to predict changed rewards. Our method imposes the similarity constraint on the spatial representations as well as the global representations, thus providing plenty of supervision for training the encoder of deep RL.

**Visual Correspondence Learning**: Visual correspondence estimation is a long-standing research in the computer vision community. It aims to establish a pair of corresponding pixels between two (or more) views taken under different locations (stereo matching) or timestep (optical flow). Recent methods for stereo matching (Žbontar & LeCun, 2016; Chang & Chen, 2018; Zhang et al., 2019) and optical flow estimation (Dosovitskiy et al., 2015; Ilg et al., 2017; Sun et al., 2018) have been advanced largely thanks to the expressive power of deep networks. Though both approaches share a similar objective of finding corresponding pixels across views, the optical flow is known to be effective for encoding temporal motion trajectories, while the stereo matching is tailored to predicting 3D depth

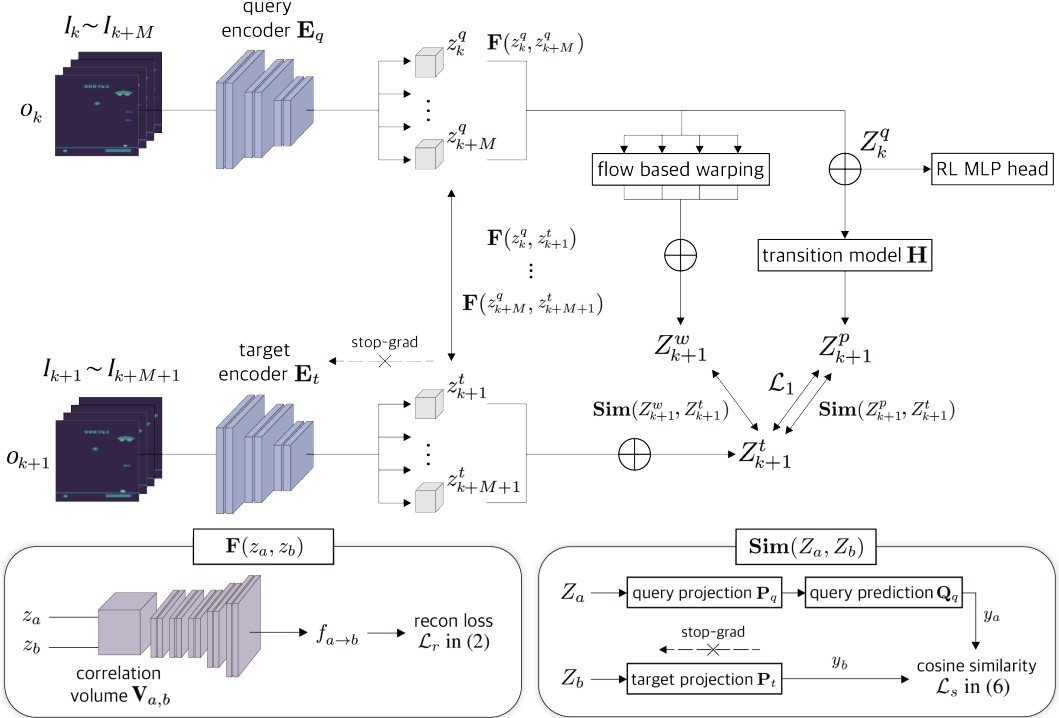

Figure 2: Overall framework of the $\mathbf{S}^3\mathbf{R}$ method: Multiple representations generated by the query and target encoders are used to infer a set of flow maps from $\mathbf{F}$. The warped representation $Z_{k+1}^w$ is produced using an inverse warping with the set of flow maps. The transition model with an action $a_k$ also predicts the future representation $Z_{k+1}^p$, similar to Schwarzer et al. (2021). The target encoder and projection heads are updated using the stop-gradient operation as in Chen & He (2020). $\oplus$ indicates a channel-wise concatenation and $1 \times 1$ convolution. The encoder representations $Z_k^q$ are used as inputs in the RL algorithm. In our work, Rainbow DQN (van Hasselt et al., 2019) ($M = 3$) and SAC (Haarnoja et al., 2018a) ($M = 2$) are used as RL algorithms.

map in the scene. The commonly used architecture for two-frame optical flow estimation involves the feature map extraction of two frames, correlation volume computation, a series of convolutions for refinement, and flow regression. While state-of-the-arts flow estimation methods require using ground truth flow maps as an explicit supervision (Dosovitskiy et al., 2015; Ilg et al., 2017; Sun et al., 2018), some unsupervised learning approaches have attempted to infer flow maps with an image reconstruction loss for imposing the constraint that corresponding pixels should have similar intensities (Ren et al., 2017; Meister et al., 2018; Wang et al., 2018; Jonschkowski et al., 2020). In our work, we present the self-supervised flow network that learns spatial representations from the consecutive frames used in the RL algorithms.

## 3 METHOD

We consider the Markov Decision Process (MDP) setting where an agent interacts with environments in a sequence of observations, actions, and rewards. We denote $o_k$, $a_k$, and $r_k$ as the observation, the action of the agent, and the reward received at timestep $k$. Since our method is a general framework that leverages the representation learning for training the RL agent, it can be combined with any RL algorithm. Following the state-of-the-arts RL approaches (Laskin et al., 2020b; Stooke et al., 2020; Schwarzer et al., 2021) using the self-supervised learning, we adopt the Soft Actor Critic (SAC) method (Haarnoja et al., 2018a) for continuous control task in DeepMind Control Suite benchmark, and Rainbow DQN (van Hasselt et al., 2019) for discrete control task in Atari Games. The proposed self-supervised spatial representations ($\mathbf{S}^3\mathbf{R}$) are used as an auxiliary task for training RL agents.

## 3.1 SELF-SUPERVISED FLOW MODEL FOR SPATIAL REPRESENTATION

We start with how to generate the spatial representations for capturing spatial deformations from the consecutively stacked frames in a self-supervised manner. An instance used by the model-free off-policy RL algorithms (Haarnoja et al., 2018a; van Hasselt et al., 2019) is a stack of images, not a single image. Given an input raw observation $o_k = \{I_k, ..., I_{k+M}\}$ where $I_k$ is an image at timestep $k$, the spatial encoder features $e_k = \{z_k, ..., z_{k+M}\}$ are first generated by applying an encoder individually to each of the input observations $o_k$. Note that $z \in \mathbb{R}^{h \times w \times d}$ is a 3-D volume with a spatial resolution $h \times w$ and a feature dimension $d$. We apply query encoder and target encoder to $o_k$ and $o_{k+1}$, respectively, and denote the output of the query encoder $e^q$ as $z^q$, and the output of the target encoder $e^t$ as $z^t$. While the existing methods (Sermanet et al., 2018; Dwibedi et al., 2018; Laskin et al., 2020b; Schwarzer et al., 2021) feeds the stacked frames to the encoder at once, which can be viewed as an early fusion (Karpathy et al., 2014), our method generates the set of the spatial latent representations individually with the encoder. Later, they are fused using $1 \times 1$ convolutional layer in a manner similar to late fusion (Simonyan & Zisserman, 2014) (see Figure 2). A similar strategy was also adopted in Shang et al. (2021) for encoding the temporal information when training the RL agent.

The set of spatial representations is used to predict the spatial deformations, *i.e.*, flow maps between two consecutive frames via a supervision by the image reconstruction loss (Godard et al., 2017; 2019; Wang et al., 2020). We compute a correlation volume $\mathbf{V}_{a,b} \in \mathbb{R}^{h \times w \times r^2}$ using a dot product between two latent representations $z_a$ and $z_b$ (Dosovitskiy et al., 2015) as follows:

$$\mathbf{V}_{a,b}(p, q) = \sum_{o \in [-\bar{r}, \bar{r}] \times [-\bar{r}, \bar{r}]} < z_a(p + o), z_b(q + o) >, \tag{1}$$

where $p$ and $q$ represent 2D feature position in $z_a$ and $z_b$, and $\bar{r}$ indicates the kernal size for computing correlation, $r = 2\bar{r} + 1$. Computing the patch similarity in (1) for all combinations of $p$ and $q$ (totally, $h^2 \cdot w^2$ times) yields huge amount of computations. Thus, the maximum displacement for computing the patch similarity is limited for $q \in N(p)$ where $N(p)$ represents a square window of size $r^2$ centered at $p$, following visual correspondence literature (Dosovitskiy et al., 2015; Ilg et al., 2017; Sun et al., 2018).

The correlation volume is fed into a series of convolutions followed by the refinement module, producing a flow map $f_{a \to b} \in \mathbb{R}^{h \times w \times 2}$ from $I_a$ to $I_b$. We follow the architecture of 'FlowNetCorr' in Dosovitskiy et al. (2015) with some modifications in the spatial resolution, encoder feature dimension and search range $r^2$ in order to keep the model complexity low. Note that while 'FlowNetCorr' is trained with the supervision by ground truth flow maps, our method relies on the image reconstruction loss for self-supervised learning such that

$$\mathcal{L}_r(f_{a \to b}) = \sum_p |I_a(p) - I_b(p + f_{a \to b})| + \mathcal{L}_{reg}, \tag{2}$$

where $I(p)$ indicates an intensity at the pixel corresponding to 2D feature position $p$. For computing the loss $\mathcal{L}_r$, we resize $I_a$ and $I_b$ to the size of the spatial representations, $h \times w$. We additionally use the Charbonnier regularization loss $\mathcal{L}_{reg}$ (Barron, 2019) for producing spatially smooth flow maps. In Figure 2, we denote $\mathbf{F}(z_a, z_b) = f_{a \to b}$ as the self-supervised flow estimation network including the correlation volume computation, the series of convolutions, and the refinement module.

## 3.2 REPRESENTATION LEARNING WITH WARPING AND SELF-PREDICTION

Figure 2 illustrates the overall architecture of the proposed $\mathbf{S}^3\mathbf{R}$ approach. Following the prior work on the self-supervised learning (He et al., 2020; Grill et al., 2020; Chen & He, 2020), we use the query encoder $\mathbf{E}_q$ with the parameters $\theta_q$ and the target encoder $\mathbf{E}_t$ with the parameters $\theta_t$ for encoding the query observation $o_k$ and the target observation $o_{k+1}$. While the parameters $\theta_q$ of the query encoder are updated through back-propagation, the parameters $\theta_t$ of the target encoder are updated with the query encoder parameters $\theta_q$ using a stop-gradient operation (Chen & He, 2020) as $\theta_t \leftarrow \theta_q$.

**Flow Learning and Warping**: By minimizing (2), we first compute a set of $M + 1$ *external* flow maps $\{f_{k+i+1 \to k+i}^{ext} | i = 0, ..., m\}$ with the self-supervised flow network $\mathbf{F}$ such that

$$f_{k+i+1 \to k+i}^{ext} = \mathbf{F}(z_{k+i+1}^t, z_{k+i}^q) \qquad \text{for } i = 0, ..., M. \tag{3}$$

Note that the external flow map is predicted from the target feature $z^t_{k+i+1}$ to the query feature $z^q_{k+i}$. Then, we warp the query features $e^q_k = \{z^q_k, ..., z^q_{k+M}\}$ into the future state via the inverse warping (Jaderberg et al., 2015) using $M + 1$ external flow maps. The warped query features $\{z^w_{k+1}, ..., z^w_{k+M+1}\}$ are then fused using $1 \times 1$ convolution, producing the warped query representation $Z^w_{k+1}$ at the timestep $k + 1$ (see Figure 2).

We can also predict *internal* flow maps within the query features $e^q_k = \{z^q_k, ..., z^q_{k+M}\}$ as $f^{int}_{a \to b} = \mathbf{F}(z^q_a, z^q_b)$. Various combinations of $a$ and $b$ are possible for computing the internal flow maps, and we choose to compute a single flow map $f^{int}_{k \to k+M}$. We found that this is an appropriate choice in terms of computational efficiency and performance as the external flow maps are already used to impose the structural similarity constraint between multiple frames, and is effective in dealing with the case where the external flow between two consecutive frames is relatively small. More details are presented in the Appendix. The loss function $\mathcal{L}_f$ for computing the internal and external flow maps is given as

$$\mathcal{L}_{flow} = \mathcal{L}_r(f^{int}_{k \to k+M}) + \sum_{i=0}^{M} \mathcal{L}_r(f^{ext}_{k+i+1 \to k+i}). \tag{4}$$

**Representation Learning with Global-level and Pixel-level Similarities**: To measure the global-level similarity between the warped query representation $Z^w_{k+1}$ and the target representation $Z^t_{k+1}$ which is the fusion of target encoder features $e^t_{k+1} = \{z^t_{k+1}, ..., z^t_{k+M+1}\}$, we use two projection heads and one predictor in a manner similar to Grill et al. (2020); Chen & He (2020). We project the two representations $Z^w_{k+1}$ and $Z^t_{k+1}$ into a smaller latent space by passing them into the query projection head $\mathbf{P}_q$ with parameters $\xi_q$ and the target projection head $\mathbf{P}_t$ with parameters $\xi_t$, and also apply an additional query prediction head $\mathbf{Q}_q$ to the query projection. The target projection head parameters $\xi_t$ are updated using the query projection head parameters $\xi_q$ with the stop-gradient operation as in the target encoder, *i.e.*, $\xi_t \leftarrow \xi_q$. The prediction loss is computed using the cosine similarity between the warped query representation $y^w_{k+1} = \mathbf{Q}_q(\mathbf{P}_q(Z^w_{k+1}))$ and the observed target representation $y^t_{k+1} = \mathbf{P}_t(Z^t_{k+1})$, which is defined as

$$\mathcal{L}_s(y^w_{k+1}, y^t_{k+1}) = -\frac{<y^w_{k+1}, y^t_{k+1}>}{\|y^w_{k+1}\|_2 \|y^t_{k+1}\|_2}. \tag{5}$$

In Figure 2, we denote the module consisting of the query projection and prediction heads and the target projection heads as '$\mathbf{Sim}(Z_a, Z_b)$' where $Z_a$ and $Z_b$ can be the warped representation from query and spatial representations from target network, respectively.

We further extend our method by leveraging a self-prediction module conditioned on an action. The action-conditioned convolutional transition model of Schwarzer et al. (2021) is applied in our framework with some modifications. We generate the query representation $Z^q_k$ by applying $1 \times 1$ convolution to the query features $\{z^q_k, ..., z^q_{k+M}\}$ and then feed it into the convolutional transition model $\mathbf{H}$. Note that we use only a single next prediction $Z^p_{k+1} = \mathbf{H}(Z^q_k, a_k)$ of the transition from the query representation $Z^q_k$, unlike Schwarzer et al. (2021) that recursively generates a sequence of $L$ predictions $Z^p_{k+1:k+L}$ with $Z^p_{k+l+1} = \mathbf{H}(Z^p_{k+l}, a_{k+l})$ for $l = 0, ..., L - 1$ and $Z^p_k = Z^q_k$. We found that since the spatial representation learning using the flow estimation and warping provide a sufficient amount of supervision, the single self-prediction is enough to train the query encoder. The self-prediction $Z^p_{k+1}$ is fed into the query projection head $\mathbf{P}_q$ and the query prediction head $\mathbf{Q}_q$ such that $y^p_{k+1} = \mathbf{Q}_q(\mathbf{P}_q(Z^p_{k+1}))$. The prediction loss is also computed using the cosine similarity loss $\mathcal{L}_s(y^p_{k+1}, y^t_{k+1})$.

The loss function $L_{sim}$ for measuring the similarity between the three representations $y^w_{k+1}, y^p_{k+1}$, and $y^t_{k+1}$ is summarized as

$$\mathcal{L}_{sim} = \mathcal{L}_s(y^w_{k+1}, y^t_{k+1}) + \mathcal{L}_s(y^p_{k+1}, y^t_{k+1}) + \lambda \mathcal{L}_1(Z^p_{k+1}, Z^t_{k+1}), \tag{6}$$

where $y^w_{k+1} = \mathbf{Q}_q(\mathbf{P}_q(Z^w_{k+1}))$, $y^p_{k+1} = \mathbf{Q}_q(\mathbf{P}_q(Z^p_{k+1}))$, $y^t_{k+1} = \mathbf{P}_t(Z^t_{k+1})$, and $\lambda$ is a hyperparameter of loss functions. We also include $L_1$ loss $\mathcal{L}_1(Z^p_{k+1}, Z^t_{k+1})$ on the original spatial latent space to impose the pixel-level similarity. Note that $L_1$ loss between $Z^w_{k+1}$ and $Z^t_{k+1}$ is not used, as the external flow loss $\mathcal{L}_r(f^{ext})$ implicitly considers the similarity between the two.

---

**Algorithm 1:** Self-Supervised Spatial ($\mathbf{S}^3$R) Representations

---

$\mathbf{E}_q$, $\mathbf{E}_t$: Query encoder, Target encoder     $\mathbf{F}$, $\mathbf{H}$: Flow model, Transition model
$\mathbf{P}_q$, $\mathbf{Q}_q$, $\mathbf{P}_t$: Query projection head, Query prediction, Target projection head
Initialize replay buffer and parameters of $\mathbf{E}_q$, $\mathbf{E}_t$, $\mathbf{P}_q$, $\mathbf{Q}_q$, $\mathbf{P}_t$, $\mathbf{F}$, $\mathbf{H}$;
**while** *Training* **do**
    • **Spatial Representation with Flow Learning**
    Generate $z_{k+i}^q$, $z_{k+i+1}^t$ with $\mathbf{E}_h$, $\mathbf{E}_g$ for $i = 0, ..., M$
    Learn external flow and internal flow with (4)
    • **Representation Learning with Warping and Self-Prediction**
    Warp a set of query features $z_{k+i}^q$ with external flow $f_{k+i+1 \to k+i}^{ext}$ for $i = 0, ..., M$.
    Generate warped query representation $Z_{k+1}^w$ by fusing a set of warped query features.
    Generate query representation $Z_k^q$ by fusing a set of query features $z_{k+i}^q$ for $i = 0, ..., M$.
    Generate predicted target representation $Z_{k+1}^p$ from $Z_k^q$ using transition model $\mathbf{H}$.
    Generate target representation $Z_{k+1}^t$ by fusing a set of target features $z_{k+i+1}^t$ for $i = 0, ..., M$.
    $Z_k^q$ goes into RL MLP head.

    Compute global and local similarity of (6) from $Z_{k+1}^w$, $Z_{k+1}^p$, and $Z_{k+1}^t$ using '**Sim**' in Figure 2.
    Optimize parameters of $\mathbf{E}_q$, $\mathbf{P}_q$, $\mathbf{Q}_q$, $\mathbf{F}$, $\mathbf{H}$ by minimizing $\mathcal{L}_{total} = \mathcal{L}_{flow} + \alpha\mathcal{L}_{sim} + \beta\mathcal{L}_{RL}$ in (7).
    Update parameters of $\mathbf{E}_t$ and $\mathbf{P}_t$ with $\mathbf{E}_q$ and $\mathbf{P}_q$;
**end**

---

Finally, the query representation $Z_k^q \in \mathbb{R}^{h \times w \times c}$ is fed into the deep RL algorithm, *e.g.*, $c = 64$ for DQN (van Hasselt et al., 2019) and $c = 32$ for SAC (Haarnoja et al., 2018a).

**Loss functions**: The final loss function is summarized as

$$\mathcal{L}_{total} = \mathcal{L}_{flow} + \alpha\mathcal{L}_{sim} + \beta\mathcal{L}_{RL}(Z_k^q), \tag{7}$$

where $\mathcal{L}_{RL}(Z_k^q)$ indicates the loss of the RL algorithms which use $Z_k^q$ as an input. $\alpha$ and $\beta$ are hyper-parameters that balance the loss functions. We summarize the overall method in Algorithm 1.

### 3.3 IMPLEMENTATION DETAILS

**Self-supervised Flow Model**: The input image $I_i$ is of $84 \times 84$ for Atari Games and DeepMind Control (DMControl) Suites. The query and target encoders generates $z_i^q, z_{i+1}^t \in \mathbb{R}^{7 \times 7 \times 64}$ ($i = k, ..., k+3$) for Atari Games and $z_i^q, z_{i+1}^t \in \mathbb{R}^{32 \times 32 \times 32}$ ($i = k, ..., k+2$) for DMControl Suites, respectively. The search window for computing the correlation volume $\mathbf{V}$ is $6 \times 6$ for Atari games and DMControl Suites. The correlation volume goes through $3 \times 3$ convolution layers 3 times. The decoder is then applied to provide a dense flow map. The decoder includes three un-convolutional layers, consisting of un-pooling and convolution, and the coarser flow maps and encoder feature maps are concatenated into each un-convolutional layer (Dosovitskiy et al., 2015).

**Action Conditioned Transition Model**: This basically follows the structure of Schwarzer et al. (2021), but in order to compute the structural similarity with the features of each frame, we set the input and output channel size to 256, which was originally set to 64 in Schwarzer et al. (2021). The transition model includes two convolutional layers interweaved with ReLU and batch normalization (Ioffe & Szegedy, 2015), with the current representations $Z_k^q$ and the action $a_k$ of one-hot vector taken to each location being fed to the first convolutional layer.

**Other Details**: The query and target projection heads, $\mathbf{P}_q$ and $\mathbf{P}_t$, are implemented as the multi-layer perceptron (MLP). For the query prediction head $\mathbf{Q}_q$, we reuse the first linear layer of the RL head similar to Schwarzer et al. (2021). We used $\lambda = 0.1$ in (6) and $\alpha = 5$, $\beta = 1$ in (7) to balance the weight of the losses. More details are presented in the Appendix.

## 4 EXPERIMENTAL RESULTS

### 4.1 EVALUATION ON ATARI GAMES

To compare the performance of the proposed method with state-of-the-arts, we chose Atari 2600 Games introduced in Kaiser et al. (2019); van Hasselt et al. (2019) where only 100K environment

Table 1: Quantitative evaluation with state-of-the-arts on the 26 Atari games (Kaiser et al., 2019) after 100K time steps using 10 random seeds: Numbers in bold represent $1^{st}$ ranking. $\mathbf{S}^3\mathbf{R}$ achieves the best performance on **13** out of **26** environments. We compared results with SimPLe (Kaiser et al., 2019), Data-Efficient Rainbow (DER) (van Hasselt et al., 2019), OverTrained Rainbow (OTRainbow) (Kielak, 2020), CURL (Laskin et al., 2020b), DrQ (Kostrikov et al., 2020), and SPR (Schwarzer et al., 2021).

| Game | Human | Random | Rainbow | SimPLe | DER | OTRainbow | CURL | DrQ | SPR | $\mathbf{S}^3\mathbf{R}$ |
|---|---|---|---|---|---|---|---|---|---|---|
| Alien | 7127.7 | 227.8 | 318.7 | 616.9 | 739.9 | 824.7 | 558.2 | 771.2 | 801.5 | **1030.1** |
| Amidar | 1719.5 | 5.8 | 32.5 | 88.0 | **188.6** | 82.8 | 142.1 | 102.8 | 176.3 | 114.3 |
| Assault | 742.0 | 222.4 | 231.0 | 527.2 | 431.2 | 351.9 | 600.6 | 452.4 | 571.0 | **708.3** |
| Asterix | 8503.3 | 210.0 | 243.6 | **1128.3** | 470.8 | 628.5 | 734.5 | 603.5 | 977.8 | 959.3 |
| Bank Heist | 753.1 | 14.2 | 15.55 | 34.2 | 51.0 | 182.1 | 131.6 | 168.9 | **380.9** | 95.8 |
| BattleZone | 37187.5 | 2360.0 | 2360.0 | 5184.4 | 10124.6 | 4060.6 | 14870.0 | 12954.0 | 16651.0 | **16688.0** |
| Boxing | 12.1 | 0.1 | -24.8 | 9.1 | 0.2 | 2.5 | 1.2 | 6.0 | 35.8 | **35.9** |
| Breakout | 30.5 | 1.7 | 1.2 | 16.4 | 1.9 | 9.8 | 4.9 | 16.1 | 17.1 | **17.5** |
| ChopperCommand | 7387.8 | 811.0 | 120.0 | 1246.9 | 861.8 | 1033.3 | 1058.5 | 780.3 | 974.8 | **1251.2** |
| Crazy Climber | 35829.4 | 10780.5 | 2254.5 | **62583.6** | 16185.3 | 21327.8 | 12146.5 | 20516.5 | 42923.6 | 42544.0 |
| Demon Attack | 1971.0 | 152.1 | 163.6 | 208.1 | 508.0 | 711.8 | 817.6 | **1113.4** | 545.2 | 884.0 |
| Freeway | 29.6 | 0.0 | 0.0 | 20.3 | **27.9** | 25.0 | 26.7 | 9.8 | 24.4 | 24.8 |
| Frostbite | 4334.7 | 65.2 | 60.2 | 254.7 | 866.8 | 231.6 | 1181.3 | 331.1 | **1821.5** | 776.9 |
| Gopher | 2412.5 | 257.6 | 431.2 | 771.0 | 349.5 | 778.0 | 669.3 | 636.3 | 715.2 | **920.3** |
| Hero | 30826.4 | 1027.0 | 487.0 | 2656.6 | 6857.0 | 6458.8 | 6279.3 | 3736.3 | **7019.2** | 3977.3 |
| Jamesbond | 302.8 | 29.0 | 47.4 | 125.3 | 301.6 | 112.3 | 471.0 | 236.0 | 365.4 | **471.4** |
| Kangaroo | 3035.0 | 52.0 | 0.0 | 323.1 | 779.3 | 605.4 | 872.5 | 940.6 | **3276.4** | 1580.0 |
| Krull | 2665.5 | 1598.0 | 1468.0 | 4539.9 | 2851.5 | 3277.9 | 4229.6 | 4018.1 | 3688.9 | **4958.3** |
| Kung Fu Master | 22736.3 | 258.5 | 0.0 | 17257.2 | 14346.1 | 5722.2 | 14307.8 | 9111.0 | 13192.7 | **17759.5** |
| Ms Pacman | 6951.6 | 307.3 | 67.0 | 1480.0 | 1204.1 | 941.9 | 1465.5 | 960.5 | 1313.2 | **1597.3** |
| Pong | 14.6 | -20.7 | -20.6 | **12.8** | -19.3 | 1.3 | -16.5 | -8.5 | -5.9 | -8.2 |
| Private Eye | 69571.3 | 24.9 | 0.0 | 58.3 | 97.8 | 100.0 | **218.4** | -13.6 | 124.0 | 158.0 |
| Qbert | 13455.0 | 163.9 | 123.46 | 1288.8 | 1152.9 | 509.3 | 1042.4 | 854.4 | 669.1 | **1290.3** |
| Road Runner | 7845.0 | 11.5 | 1588.46 | 5640.6 | 9600.0 | 2696.7 | 5661.0 | 8895.1 | **14220.5** | 3175.7 |
| Seaquest | 42054.7 | 68.4 | 131.69 | 683.3 | 354.1 | 286.9 | 384.5 | 301.2 | 583.1 | **734.9** |
| Up N Down | 11693.2 | 533.4 | 504.6 | 3350.3 | 2877.4 | 2847.6 | 2955.2 | 3180.8 | **28138.5** | 4263.8 |

Table 2: Quantitative evaluation of mean and standard deviation with state-of-the-arts on the DM-Control suite (Tassa et al., 2018) after 100K time steps and 500K time steps using 10 random seeds. Numbers in bold represent $1^{st}$ ranking, and $\mathbf{S}^3\mathbf{R}$ achieves the best performance on **4** out of **6** environments for 500K time steps. We compared results with state-based SAC and pixel-based SAC (Haarnoja et al., 2018b), SAC+AE (Yarats et al., 2019), Dreamer (Hafner et al., 2019a), PlaNet (Hafner et al., 2019b), CURL (Laskin et al., 2020b), RAD (Laskin et al., 2020a), and DrQ (Kostrikov et al., 2020).

| 100K step scores | State SAC | Pixel SAC | SAC+AE | Dreamer | PlaNet | CURL | RAD | DrQ | $\mathbf{S}^3\mathbf{R}$ |
|---|---|---|---|---|---|---|---|---|---|
| Finger, Spin | 811±46 | 179±66 | 740±64 | 341±70 | 136±216 | 767±56 | 856±73 | **901±104** | 880±127 |
| Cartpole, Swingup | 835±22 | 419±40 | 311±11 | 326±27 | 297±39 | 582±146 | 828±27 | 759±92 | **841±47** |
| Reacher, Easy | 746±25 | 145±30 | 274±14 | 314±155 | 20±50 | 538±233 | **826±219** | 601±213 | 621±202 |
| Cheetah, Run | 616±18 | 197±15 | 267±24 | 235±137 | 138±88 | 299±48 | **447±88** | 344±67 | 251±34 |
| Walker, Walk | 891±82 | 42±12 | 394±22 | 277±12 | 224±48 | 403±24 | 504±191 | **612±164** | 595±104 |
| Ball in Cup, Catch | 746±91 | 312±63 | 391±82 | 246±174 | 0±0 | 769±43 | 840±179 | 913±53 | **922±60** |

| 500K step scores | State SAC | Pixel SAC | SAC+AE | Dreamer | Planet | CURL | RAD | DrQ | Ours |
|---|---|---|---|---|---|---|---|---|---|
| Finger, Spin | 923±21 | 179±166 | 884±128 | 796±183 | 561±284 | 926±45 | 947±101 | 938±103 | **954±131** |
| Cartpole, Swingup | 848±15 | 419±40 | 735±63 | 762±27 | 475±71 | 841±45 | 863±9 | 868±10 | **880±34** |
| Reacher, Easy | 923±24 | 145±30 | 627±58 | 793±164 | 210±390 | 929±44 | **955±71** | 942±71 | 932±41 |
| Cheetah, Run | 795±30 | 197±15 | 550±34 | 570±253 | 305±131 | 518±28 | **728±71** | 660±96 | 501±63 |
| Walker, Walk | 948±54 | 42±12 | 847±48 | 897±49 | 351±58 | 902±43 | 918±16 | 921±45 | **930±75** |
| Ball in Cup, Catch | 974±33 | 312±63 | 794±58 | 879±87 | 460±380 | 959±27 | 974±12 | 963±9 | **988±54** |

steps, corresponding to two hours of gameplay experiences, are available for training data. This sample-efficient setup, which uses much less environment steps than the standard setup of 50,000K environment steps, has been adopted for evaluating the performance of recent sample-efficient deep RL algorithms (Kaiser et al., 2019; van Hasselt et al., 2019; Kielak, 2020; Laskin et al., 2020b;

Table 3: To study the impact of several losses, we measured the average performance over 10 random seeds according to the combinations of losses on DMControl Suite (Tassa et al., 2018) with 500K time steps. Refer to section 4.3 for 'F', 'F+W', 'P', and 'F+P'.

| 500K step scores | F | F+W | P | F+P | F+W+P (= $\mathbf{S}^3\mathbf{R}$) |
|---|---|---|---|---|---|
| Finger, Spin | 729±110 | 757±100 | 711±64 | 768±112 | **834±95** |
| Cartpole, Swingup | 819±38 | 876±19 | 793±15 | 868±27 | **880±34** |
| Reacher, Easy | 857±45 | 901±29 | 904±58 | 922±31 | **932±41** |
| Cheetah, Run | 366±63 | 411±53 | **482±87** | 392±69 | 448±65 |
| Walker, Walk | 770±87 | 879±67 | 822±53 | 876±41 | **914±30** |
| Ball in Cup, Catch | 849±42 | 953±26 | 848±107 | 951±20 | **962±14** |

Schwarzer et al., 2021). We compared our results with various RL algorithms including SimPLe (Kaiser et al., 2019) which learns a pixel-level transition model for Atari, Data-Efficient Rainbow (DER) (van Hasselt et al., 2019) which modifies the Rainbow hyperparameters for improving the sample efficiency, OTRainbow (Kielak, 2020) which is an over-trained version of the Rainbow for the sample efficiency, CURL (Laskin et al., 2020b) which proposes the use of image augmentation with the contrastive loss (van den Oord et al., 2018) for self-supervised representation learning, DrQ (Kostrikov et al., 2020) which uses the modest image augmentation to improve the sample efficiency, and SPR (Schwarzer et al., 2021) which trains an agent to predict its own latent state representations into the future. Following the experimental setup on the above-mentioned approaches, we evaluated on 26 environments of Atari 2600 games by measuring the average return after 100K interaction steps. As shown in Table 1, the proposed method ($\mathbf{S}^3\mathbf{R}$) achieved the best performance on 13 out of 26 environments. We trained our method with 10 random seeds, similar to other methods.

## 4.2 EVALUATION ON DMCONTROL SUITE

Various approaches including ours have been benchmarked on the DMControl Suite where the agent operates from pixels to evaluate challenging visual continuous control tasks (Tassa et al., 2018). We compared our results with State-SAC which supposes that the agent has access to low-level state based features, Pixel-SAC (Haarnoja et al., 2018b) which directly operates from pixels, SAC+AE (Yarats et al., 2019) which uses a joint learning of SAC with $\beta$-VAE (Higgins et al., 2017), VAE (Kingma & Welling, 2013), and regularized autoencoder (Vincent et al., 2008), Dreamer (Hafner et al., 2019a) and PlaNet (Hafner et al., 2019b) which learn a latent space world model, CURL (Laskin et al., 2020b) which uses image augmentation with the contrastive loss (van den Oord et al., 2018), RAD (Laskin et al., 2020a) and DrQ (Kostrikov et al., 2020) which demonstrate that data augmentation can greatly improve the performance of model-free RL algorithms and achieve state-of-the-art performance on DMControl Suite. Table 2 demonstrates that the self-supervised spatial representations of $\mathbf{S}^3\mathbf{R}$ achieved best performance on **4** out of **6** environments for 500K time steps including Cartpole Swingup, Reacher Easy, Walker Walk and Ball in Cup Catch. We trained our method with 10 random seeds, and the results with 5 random seeds are provided in the Appendix.

## 4.3 ABLATION STUDY

Table 3 measured the average performance over 10 random seeds according to the combinations of several losses on DMControl Suite (Tassa et al., 2018) with 500K time steps.
• 'F' using only the flow loss in (4)
• 'F+W' using flow loss and similarity loss with 'W'arped query and target representations in (6)
• 'P' using prediction loss with self-'P'redicted and target representations in (6)
• 'F+P' using the flow loss, L1 loss, and the prediction loss in (6).
The network trained with only 'F' produces worse performance compared to 'F+W', 'F+P' and 'F+W+P', but still produces comparable performance to state-of-the-arts, implying that even without the warping and transition model, simply guiding the encoder to extract features for the flow prediction helps the RL agent to perform well enough. The performance of 'F+W' and 'F+P' is similar, but 'F+W' has slightly better performance on step scores with smaller standard deviations. This implies that the warped query representation to the future state using the estimated flow is capable of providing as useful supervision as the self-predicted representation that uses an action-conditioned transition model to predict future representations. The performance was further boosted, when using 'F+W+P' altogether ($= \mathbf{S}^3\mathbf{R}$). To measure only the impact of each loss, data augmentation was not performed. Note that 'P' is similar to SPR method (Schwarzer et al., 2021) using only one prediction.

## 5 CONCLUSION

We have presented the self-supervised spatial representations, termed $\mathbf{S}^3\mathbf{R}$ to encode local spatial structures in an unsupervised manner. The flow maps inferred by the proposed method offer plenty of supervision for learning the spatial representations, and also compute warped predictions at future frame. $\mathbf{S}^3\mathbf{R}$ achieves state-of-the-art performance on Atari benchmark with 100K steps and DMControl Suites with 100K/500K steps. We have shown the importance of learning the spatial representations in improving the performance and sample-efficiency of image-based RL algorithms. We hope this can facilitate future works at various aspects for RL based on self-supervised learning.

## 6 REPRODUCIBILITY STATEMENT

For the reproducibility of the proposed method, we opened our code at `https://sites.google.com/view/iclr2022-s3r` as stated in the Abstract. We also provide the full hyperparameters in Section A of Appendix, so others can use the code with the same settings we experimented with.

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

## A IMPLEMENTATION DETAILS

This section provides tables summarizing the hyperparameters used for experiments on Atari Games (Kaiser et al., 2019) and DMControl Suite (Tassa et al., 2018) and the detailed description of our network architecture.

### A.1 HYPERPARAMETERS ON ATARI GAMES AND DMCONTROL SUITE

Table 4: Hyperparameters used for $\mathbf{S}^3\mathbf{R}$ experiments on Atari Games

| Parameter | Value |
|---|---|
| Observation Size | (84, 84) |
| Augmentation | Random shifts ($\pm$4 pixels), Intensity (scale=0.05) |
| Image Gray-scale | True |
| Update | Distributional Q |
| Stacked Frames | 4 |
| Action Repeat | 4 |
| Reward Clipping | [-1, 1] |
| Training Steps | 100K |
| Evaluation Trajectories | 100 |
| Minimum Replay Size (for sampling) | 2000 |
| Max Frames (per episode) | 108K |
| Support Of Q-Distribution | 51 bins |
| Discount Factor | 0.99 |
| Optimizer | Adam |
| Optimizer: learning rate | 0.0001 |
| Optimizer: $\beta_1$ | 0.9 |
| Optimizer: $\beta_2$ | 0.999 |
| Optimizer: $\epsilon$ | 0.00015 |
| Max Gradient Norm | 10 |
| Multi Step Return | 10 |
| Target Network: Update Period | 1 |
| Q Network: Channels | 32, 64, 64 |
| Q Network: Filter Size | $8 \times 8, 4 \times 4, 3 \times 3$ |
| Q Network: Stride | (4, 4), (2, 2), (1, 1) |
| Q Network: Hidden Units | 256 |
| Non-Linearity | ReLU |
| Replay Period Every | 1 |
| Updates Per Step | 2 |
| Exploration | Noisy Nets |
| Noisy Nets Parameter | 0.5 |
| Priority Exponent | 0.5 |
| Priority Correction | $0.4 \rightarrow 1$ |

For the reproducibility of our work, we provide full hyperparameters for our experiments on Atari Games (Kaiser et al., 2019) in Table 4. We follow the common practices used to set up Rainbow DQN van Hasselt et al. (2019) in existing methods Laskin et al. (2020b); Schwarzer et al. (2021) for experimenting on Atari Games. Table 5 shows a full list of hyperparameters for DMControl suites experiments. We utilize the similar hyperparameters and optimizer to CURL Laskin et al. (2020b).

Table 5: Hyperparameters used for $\mathbf{S}^3\mathrm{R}$ experiments on DMControl

| Parameter | Value |
|---|---|
| Observation Size | (84, 84) |
| Observation Rendering | (100, 100) |
| Augmentation | Random crop, translation |
| Stacked Frames | 3 |
| Action Repeat | 2 (finger-spin, walker-walk), 8 (cartpole-swingup), 4 (otherwise) |
| Evaluation Episodes | 10 |
| Discount Factor | 0.99 |
| Optimizer | Adam |
| $(\beta_1, \beta_2) \rightarrow (f_\theta, \pi_\psi, Q_\phi)$ | (0.9, 0.999) |
| $(\beta_1, \beta_2) \rightarrow (\alpha)$ | (0.5, 0.999) |
| Learning Rate $(f_\theta, \pi_\psi, Q_\phi)$ | $2e-4$ (cheetah-run), $1e-3$ (otherwise) |
| Learning Rate $(\alpha)$ | $1e-4$ |
| Batch Size | 64 |
| Replay Buffer Size | 100000 |
| Initial Steps | 1000 |
| Hidden Units (MLP) | 1024 |
| Q Function EMA $\tau$ | 0.01 |
| Critic Target Update Frequency | 2 |
| Convolutional Layers | 4 |
| Number Of Filters | 32 |
| Non-Linearity | ReLU |
| Latent Dimension | 50 |
| Initial Temperature | 0.1 |

## A.2 NETWORK ARCHITECTURE

The detailed network architecture of our method is provided in Table 6. We describe the architectures of the encoder, FlowNet and transition model. 'N', 'K' and 'S' of convolution operations represent the channel, kernel size, and stride, respectively. 'LReLU' and 'BN' indicate LeakyReLU Maas et al. (2013) and Batch Normalization Ioffe & Szegedy (2015). We also provide input and output tensors of each layer for better understanding.

Table 6: Detailed describtion of the proposed network architecture

| | Encoder | | |
|---|---|---|---|
| Layer | Operations | Input | Output |
| 1 | Conv(N32, K8, S4) - ReLU - DropOut | $\mathbf{I_t} \sim \mathbf{I_{t+M}}$ | $en1_t \sim en1_{t+M}$ |
| 2 | Conv(N64, K4, S2) - ReLU - DropOut | $en1_t \sim en1_{t+M}$ | $en2_t \sim en2_{t+M}$ |
| 3 | Conv(N64, K3, S1) - ReLU - DropOut | $en2_t \sim en2_{t+M}$ | $en3_t \sim en3_{t+M}$ |
| | FlowNet | | |
| Layer | Operations | Input | Output |
| 4 | Conv(N32, K1, S1) - BN - LReLU | $en3_t$ | $\tilde{en}3_t$ |
| 5 | Compute Correlation Volume | $en3_t, en3_{t+k}$ | corr |
| 6 | Concatenation | $\tilde{en}3_t$, corr | conc0 |
| 5 | Conv(N256, K3, S1) - BN - LReLU | conc0 | conv1 |
| 6 | Conv(N512, K3, S1) - BN - LReLU | conv1 | conv2p |
| 7 | Conv(N512, K3, S1) - BN - LReLU | conv2p | conv2 |
| 8 | Conv(N512, K3, S1) - BN - LReLU | conv2 | conv3p |
| 9 | Conv(N512, K3, S1) - BN - LReLU | conv3p | conv3 |
| 10 | Conv(N2, K3, S1) | conv3 | flow3 |
| 11 | Upsampling | flow3, conv2 | flow3up, conv2up |
| 12 | Deconv(N256, K4, S2) - LReLU | conv3 | conv3d |
| 13 | Concatenation | conv2up, conv3d, flow3up | conc3 |
| 14 | Conv(N2, K3, S1) | conc3 | flow2 |
| 15 | Upsampling | flow2, conv1 | flow2up, conv1up |
| 16 | Deconv(N128, K4, S2) - LReLU | conc3 | conc3d |
| 17 | Concatenation | conv1up, conc3d, flow2up | conc2 |
| 18 | Conv(N2, K3, S1) | conc2 | flow1 |
| 19 | Upsampling | flow1, $en2_t$ | flow1up, $en2up_t$ |
| 20 | Deconv(N130, K4, S2) - LReLU | conc2 | conc2d |
| 21 | Concatenation | $en2up_t$, conc2d, flow1up | conc1 |
| 22 | Conv(N2, K3, S1) | conc1 | flow0 |
| | Transition Model | | |
| Layer | Operations | Input | Output |
| 1 | Onehot Encoding (Only for Atari) | $a_k$ | $a_k$ |
| 2 | Concatenate | $Z_k^q, a_k$ | conc |
| 3 | Conv(N256, K3, S1) - ReLU | conc | conv1t |
| 4 | Conv(N64, K3, S1) - ReLU | conv1t | conv2t |
| 5 | Min-Max Normalize | conv2t | $Z_{k+1}^p$ |

## A.3 ENVIRONMENTS

$\mathbf{S}^3$R was implemented in Pytorch (Paszke et al., 2019) with the use of rlpyt (Stooke & Abbeel, 2019) and Mujoco (Todorov et al., 2012) license, and was simulated on 1 Titan RTX GPU.

A.4    DETAILS OF USING FLOW MAPS

**Use of internal flow**: When four consecutive frames are used in Rainbow DQN ($M = 3$), the external flows are specified using query images $o_k$ ($I_k \sim I_{k+3}$) and target images $o_{k+1}$ ($I_{k+1} \sim I_{k+4}$) as follows:

- flow ($z_k^q, z_{k+1}^t$)
- flow ($z_{k+1}^q, z_{k+2}^t$)
- flow ($z_{k+2}^q, z_{k+3}^t$)
- flow ($z_{k+3}^q, z_{k+4}^t$).

The internal flow is computed using query images $o_k$ as

- flow ($z_k^q, z_{k+3}^q$).

The external flows computed between consecutive two frames are used to warp the query representations. The internal flow is also predicted using the same self-supervised flow network yet with two distant frames ($z_k^q \rightarrow z_{k+3}^q$). We found that this is often effective in dealing with the case where the flow between two consecutive frames is relatively small. Namely, the internal flow between $k^{th}$ and $(k+3)^{th}$ frames can be complementary when the external flows are rather small and thus flow learning becomes less effective.

The internal flow is computed with a stack of images $o_k$, while the external flows are computed between $o_k$ and $o_{k+1}$. This is why we name two flow parts 'internal' flow and 'external' flow, respectively. Note that the internal flow is computed only between two distant frames ($k^{th}$ and $(k+3)^{th}$ frames in $o_k$), considering that the external flow is computed between two consecutive frames as above.

**Image and representation warping**: The representation we use contains spatial information. For instance, suppose the convolutional feature map $z$ of the size $32 \times 32 \times 64$ is generated from an input image $I$ of the size $128 \times 128 \times 3$. The feature map can have a lower spatial resolution (by the downsampling operator such as max-pooling or strided convolution) and a larger channel dimension than the input image. The 64 dimensional vector at a spatial position of the feature map represents the information of corresponding positions of the input image. Therefore, we can employ the flow used for image warping for warping the feature map (query representations). This kind of implementation has been commonly used in many computer vision tasks where an image alignment is required.

**Visualization of flow maps**: For a better understanding of the proposed method, the examples of the flow maps predicted by $\mathbf{S}^3$R are represented in Figure 3.

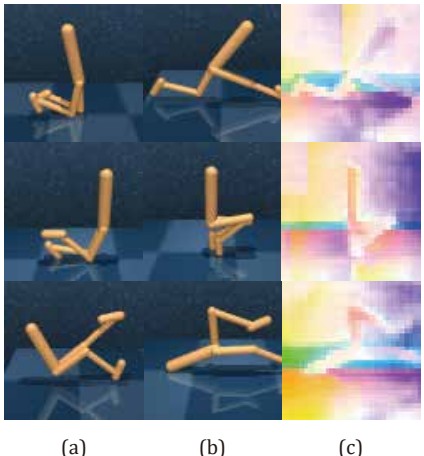 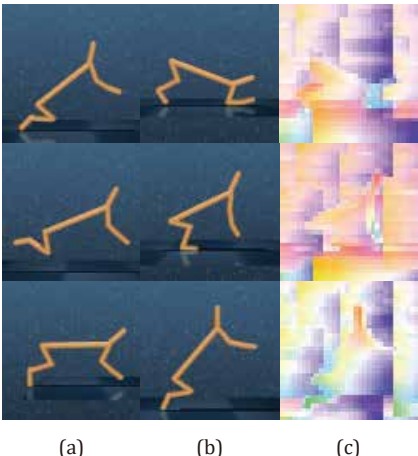

|     |     |     |     |     |     |
| (a) | (b) | (c) | (a) | (b) | (c) |

Figure 3: Visualization of the flow maps learned by $\mathbf{S}^3$R: (a) The source frame, (b) the target frame, and (c) the flow map of the task 'Walker, Walk' and 'Cheetah, Run' of the DM Control Suite.

## B    PERFORMANCE CONSISTENCY EVALUATION

We consider that the performance depends on the choice of the seed, so we measure the performance by using 10 random seeds and averaging results. To prove that our performance improvement is consistent and not caused by the noise of the estimation, we observed the change in the average performance according to the number of the random seeds. Figure 4 and 5 represent the quantitative evaluation on the Atari games (Kaiser et al., 2019) and DMControl Suite (Tassa et al., 2018) according to the number of the random seed. We show result of 5 random seeds and 10 random seeds to compare the performance. In Figure 5, we additionally showed the standard deviation of the performance. The bar graph represents the average of the performance, and the line graph represents the standard deviation of the performance.

From Figure 4 and 5, it can be seen that our results do not differ significantly depending on the number of random seeds. It can be seen that our performance is significantly higher than the SOTA performance regardless of random seed, and is consistent.

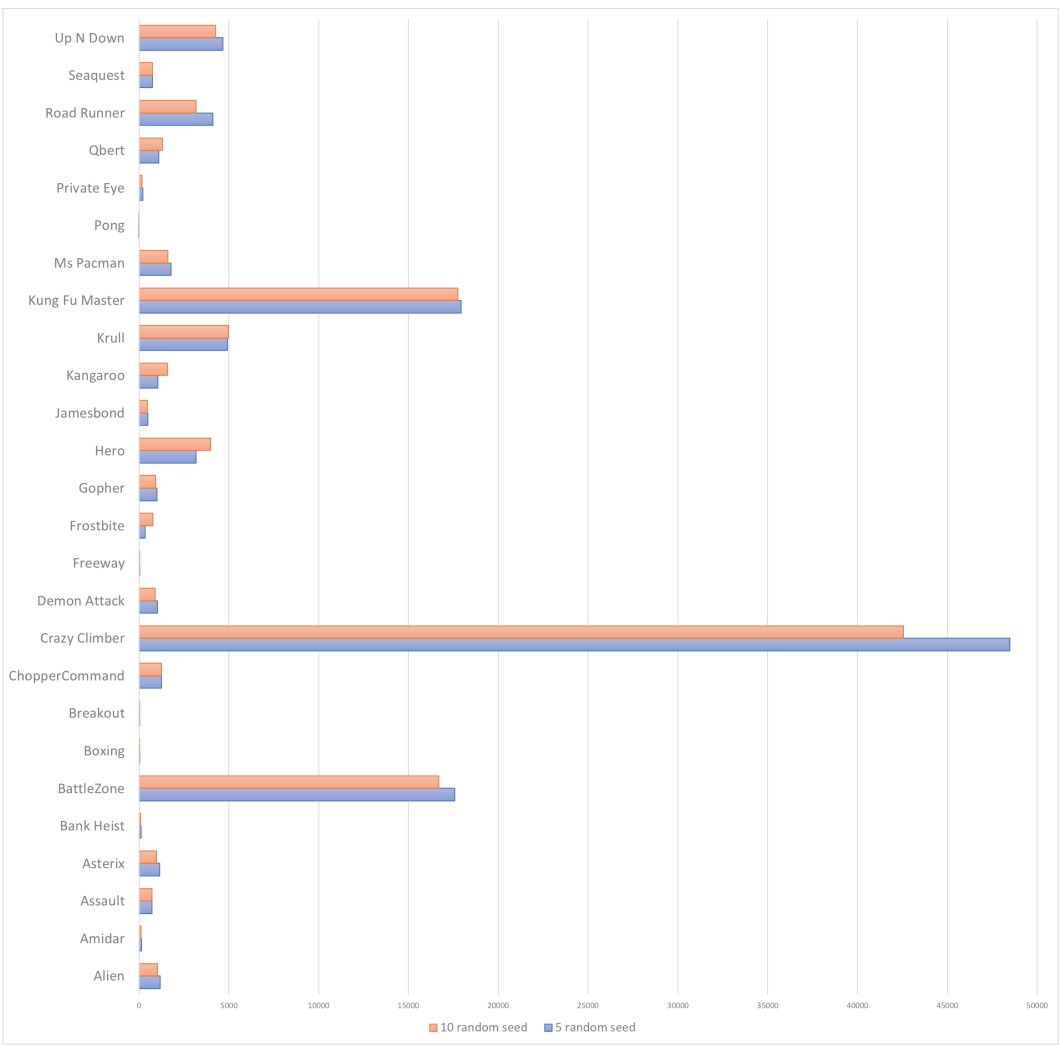

Figure 4: Quantitative evaluation on the 26 Atari games (Kaiser et al., 2019) according to the number of the random seeds: We show the mean performance on the 26 Atari games (Kaiser et al., 2019) after 100K time steps for each 5 random seeds and 10 random seeds to show the consistency of the proposed method regardless of random seeds.

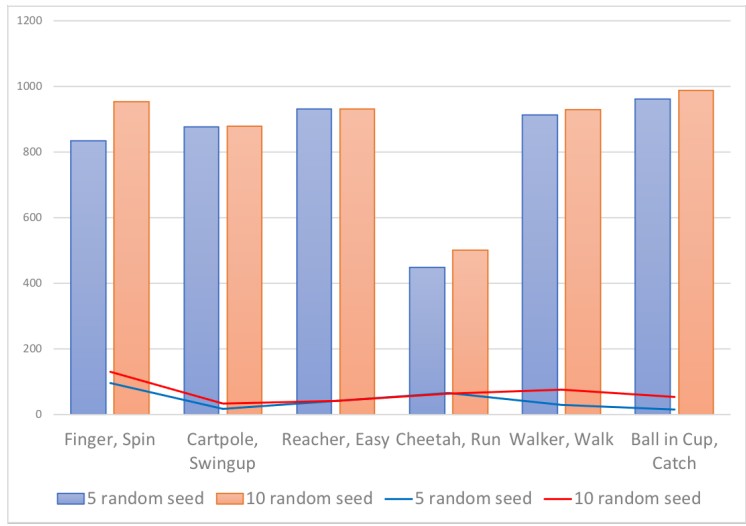

Figure 5: Quantitative evaluation on the DMControl Suite (Tassa et al., 2018) according to the number of the random seed: We show the mean performance on the DMControl Suite (Tassa et al., 2018) after 500K time steps for each 5 random seeds and 10 random seeds to show the consistency of the proposed method regardless of random seed.

## C   ABLATION STUDY ON DATA AUGMENTATION

As described in RAD (Laskin et al., 2020a) and DrQ (Kostrikov et al., 2020), image augmentation can improve the data-efficiency and generalization of RL methods. To study the impact of data augmentation when used with the proposed method, we measured the average performance over 10 random seeds according to the data augmentation on DMControl Suite (Tassa et al., 2018). In Table 7, we evaluated the performance of the proposed method when used with crop and translation proposed in Laskin et al. (2020a).

Slightly different from the result presented in Laskin et al. (2020a), Cartpole Swingup and Reacher Easy has the best performance when no augmentation was used, Finger Spin and Cheetach Run has the best performance for translation, and Walker Walk and Ball in cup Catch has the best performance for crop. Since $S^3R$ learns flow in an end-to-end manner with RL algorithm, it is analyzed that the results are different from those of Laskin et al. (2020a).

Table 7: To study the impact of various data augmentation, we measured the average performance over 10 random seeds according to the data augmentation on DMControl Suite (Tassa et al., 2018) with 500K time steps.

| 500K step scores | $S^3R$ + no aug | $S^3R$ + crop | $S^3R$ + translation |
|---|---|---|---|
| Finger, Spin | 834±95 | 821±128 | **954±131** |
| Cartpole, Swingup | **880±34** | 837±16 | 872±51 |
| Reacher, Easy | **932±41** | 833±87 | 908±79 |
| Cheetah, Run | 448±65 | 412±81 | **501±63** |
| Walker, Walk | 914±30 | **930±75** | 886±51 |
| Ball in cup, Catch | 962±14 | **988±54** | 946±42 |

# D    ADDITIONAL EVALUATION METRIC

In all experiments, the evaluation on Atari Games (Kaiser et al., 2019) was conducted by measuring the performance with 10 or more random seeds, following the previous studies including SPR (Schwarzer et al., 2021) and CURL (Laskin et al., 2020b). Recently, Agarwal et al. (2021) analyzes the problems related to statistical uncertainty in the existing evaluation method of Atari Games (Kaiser et al., 2019). Accordingly, we add the results of a more complete evaluation using 'Probability of Improvement' proposed in Agarwal et al. (2021) in Figure 6.

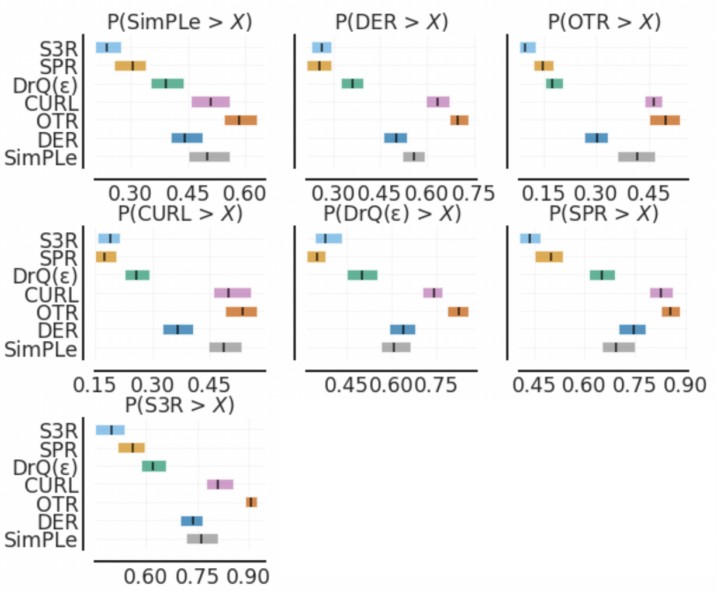

Figure 6: Evaluation result of 'Probability of Improvement' proposed in Agarwal et al. (2021).

This evaluation method estimates how likely an algorithm improves upon another algorithm Agarwal et al. (2021). For instance, 'P(SimPLe>X)' indicates the probability (written in the horizontal line) that 'SimPLe' is better than another algorithm, called 'X', which is listed in the vertical line (e.g., S3R, SPR,..., DER). Namely, the probability that SimPLe is better than SPR, 'P(SimPLe>SPR)', has an average value of about 0.3.

In this context, in the six graphs above, the smaller the value of $\mathbf{S}^3\mathbf{R}$, the higher the performance. Also, in the graph of $\mathbf{S}^3\mathbf{R}$ below, 'P(S3R > X)', $\mathbf{S}^3\mathbf{R}$ ranks the highest as the remaining bars of algorithms are located to the right of the bar of $\mathbf{S}^3\mathbf{R}$. From this analysis, it can be reconfirmed that $\mathbf{S}^3\mathbf{R}$ has the most superior performance compared to the state-of-the-art methods on Atari Games (Kaiser et al., 2019) as mentioned in the result of Section 4.1.

# E  ANALYSIS ON COMPUTATIONAL COST AND PERFORMANCE IMPROVEMENT

**Computational cost**: The increase in the computational cost for training is unavoidable because $\mathbf{S}^3\mathbf{R}$ additionally leverage the flow prediction and warping networks used in the vision task, but we found that the additional computational cost for training is not so significant. For training on DMControl Suite (Tassa et al., 2018) up to 500K on the same GPU environment, the proposed method takes about 16 hours, whereas the state-of-the-art methods CURL (Laskin et al., 2020b) and SPR (Schwarzer et al., 2021) take about 10 hours and 13 hours, respectively. Note that the original SPR paper did not provide the code implemented for DM Control Suite, so we conducted the experiments by modifying the original SPR code. Additionally, the flow prediction and warping networks are used only during training, and the inference process is implemented in the same manner as other methods. Therefore, the inference time of our method is exactly the same as that of the state-of-the-arts methods (CURL (Laskin et al., 2020b), SPR (Schwarzer et al., 2021), DrQ (Kostrikov et al., 2020)) as long as the same encoder for query images is used.

We further analyzed the performance improvement by our method based on the two benchmarks used in Section 4.

**26 Atari Games**: Based on the evaluation method used in the existing Atari Games (Kaiser et al., 2019), CURL (Laskin et al., 2020b) recorded the highest mean in 7 games out of 26, and SPR (Schwarzer et al., 2021) recorded the highest mean in 11 games out of 26. $\mathbf{S}^3\mathbf{R}$ has the highest mean in 13 games out of 26. It can be interpreted that the performance increase of the $\mathbf{S}^3\mathbf{R}$ is not small by considering the quantitative aspects of these games. Also, among the 13 games in which $\mathbf{S}^3\mathbf{R}$ has an edge, in particular, in 8 games (Alien, Assault, Gopher, Jamesbond, Krull, Kung Fu Master, Ms Pacman, and Seaquest), $\mathbf{S}^3\mathbf{R}$ records a remarkably higher performance compared to other methods. Even the performance of certain games is high enough to match that of humans. This is because the proposed method of capturing the local spatial structure is able to derive an effective representation from the images of the specific Atari Games with various movements.

However, $\mathbf{S}^3\mathbf{R}$ may not be effective for some games. In particular, $\mathbf{S}^3\mathbf{R}$ did not perform well in the task 'Pong' in Atari Games (Kaiser et al., 2019). The biggest reason for this is that there are too few discriminative spatial structures available in the game images. Therefore, we can be sure that our representation learning method, which effectively captures the spatial structure, will work particularly well for data with much more complex structural features. In other words, while most of the simple methods suffer from training with data with complex structural features, $\mathbf{S}^3\mathbf{R}$ can be a good substitute for addressing this.

A method with relatively low-complexity may be preferred depending on the situation. However, as mentioned above, it was shown from the Atari benchmark that it is a much better choice to use $\mathbf{S}^3\mathbf{R}$ in the tasks that need to capture complex structural features.

**DMControl Suite**: In the case of DMControl Suite (Tassa et al., 2018), the performance at 100K steps is usually based on when most methods do not converge. In Table 2, the performance of S3R recorded the highest mean in 2 tasks out of 6 tasks, and RAD (Laskin et al., 2020a) and DrQ (Kostrikov et al., 2020) also recorded the highest mean in 2 tasks out of 6 tasks, respectively. It can be interpreted that RAD (Laskin et al., 2020a), DrQ (Kostrikov et al., 2020) and $\mathbf{S}^3\mathbf{R}$ are the three methods with the highest convergence speed.

In general, the performance at 500K steps after most methods converge is widely adopted for the evaluation. In Table 2, $\mathbf{S}^3\mathbf{R}$ shows the highest performance in 4 out of 6 tasks compared to state-of-the-arts. Considering that this performance is the average value obtained by running 10 random seeds, it is a credible assessment. Also, when compared to the performance improvement rate of other methods, the performance increase of S3R is by no means small.

