# OpenReview forum: "Self-Supervised Structured Representations for Deep Reinforcement Learning"
_ICLR.cc/2022/Conference — ICLR 2022 Submitted_

### Official Review · Reviewer_iodY · 2021-10-24

**Correctness:** 4
**Technical Novelty And Significance:** 4
**Empirical Novelty And Significance:** 2
**Recommendation:** 8
**Confidence:** 2

**Main Review:**

The motivation is justified because indeed previous literature has shown that when humans play Atari games, representation of objects are important for efficient learning. Learning global representation may lose detailed information of objects. Although the current work does not attempt to explicitly model objects, the flow information likely captures some object information implicitly.
I find the results generally convincing and the idea appears novel and interesting.
However, I am a bit worried why the internal flow loss in equation 4 will help the network extract useful information. If my understanding is correct, the flow for warping internal representation is the same flow as the one for warping image (equation 2). How can we guarantee this the extracted query feature does not simply become a fixed linear transformation of the local image patch? As long as the representation in equation 1 can be learned to predict external flow that satisfies the image warping cost, it seems to be able to guarantee the cost in internal flow to be low as well, even if query feature is a simple linear feature of local image patch.
Although the paper has demonstrated good performance in the RL task, it may be interesting, if possible, to provide some visualization of what kind of local feature is actually learned, in the appendix.

**Summary Of The Paper:**

The paper proposes a new self-supervised representational learning for reinforcement learning tasks, and demonstrated superior performance in the sample-efficient setting against states-of-the-arts models. The major idea is to force neural networks to encode optical flow between consecutive frames and impose regularization that the local embedding predicted by warping and a state transition model (given action) are both similar to the extracted embedding from the next frame.

**Summary Of The Review:**

I think the paper provides an interesting new objective for encouraging the representation by self-supervised learning also captures important local information for efficient learning of RL tasks.

---

> ### Author Response · Authors · 2021-11-21
> **Response**
>
> We highly appreciate your positive review and valuable feedback.
> Thank you very much for pointing out the parts that needs further explanation.
>
> **Q1. Why the internal flow loss in equation 4 will help the network extract useful information? As long as the representation in equation 1 can be learned to predict external flow that satisfies the image warping cost, it seems to be able to guarantee the cost in internal flow to be low as well, even if query feature is a simple linear feature of local image patch.**
>
> : When four consecutive frames are used in Rainbow DQN (M=3), the external flows are specified using query images $o_k$ ($I_k$ ~ $I_{k+3}$) and target images $o_{k+1}$ ($I_{k+1}$ ~ $I_{k+4}$) as follows:
> - flow ($z^q_k$, $z^t_{k+1}$)
> - flow ($z^q_{k+1}$, $z^t_{k+2}$)
> - flow ($z^q_{k+2}$, $z^t_{k+3}$)
> - flow ($z^q_{k+3}$, $z^t_{k+4}$).
>
> The internal flow is computed using query images $o_k$ as
> - flow ($z^q_k$, $z^q_{k+3}$).
>
> The external flows computed between consecutive two frames are used to warp the query representations. As you mentioned, the internal flow is also predicted using the same self-supervised flow network yet with two distant frames ($z^q_k$ $\rightarrow$ $z^q_{k+3}$). We found that this is often effective in dealing with the case where the flow between two consecutive frames is relatively small. Namely, the internal flow between $k^{th}$ and ${(k+3)}^{th}$ frames can be complementary when the external flows are rather small and thus flow learning becomes rather less effective.
>
> The internal flow is computed with a stack of images $o_k$, while the external flow is computed between $o_k$ and $o_{k+1}$. This is why we name two flow parts "internal" flow and "external" flow, respectively. Note that the internal flow is computed only between two distant frames ($k^{th}$ and ${(k+3)}^{th}$ frames in $o_k$), considering that the external flow is computed between two consecutive frames as above. We will clarify this in the paper.
>
> **Q2. The flow for warping internal representation is the same flow as the one for warping image. How can we guarantee this the extracted query feature does not simply become a fixed linear transformation of the local image patch?**
>
> : We use the term "structured representation" with the intention to indicate 3D volume which consists of 2D spatial coordinate and 1D feature dimension. Namely, the representation we use contains spatial information. For instance, suppose the convolutional feature map z of the size $32 \times 32 \times 64$ is generated from an input image I of the size $128 \times 128 \times 3$. The feature map can have a lower spatial resolution (by the downsampling operator such as max-pooling or strided convolution) and a larger channel dimension than the input image. The 64 dimensional vector at a spatial position of the feature map represents the information of corresponding positions of the input image.
>
> Therefore, we can use the flow used for image warping for warping the feature map (query/target representation). Actually, this kind of implementation has been commonly used in many computer vision tasks where an image alignment is required. We will provide the above detailed explanation in the revised version of the paper.
>
> Also, as you advised, attaching an intermediate visualizations (i.e. representation, flow map) to Appendix will help readers to better understand. We add a visualization like the figure below to the Appendix of the revised paper.
>
> [Figure](https://drive.google.com/file/d/1eeGFI_nsedtYafFrMS4bhGTEZ3yuOuec/view?usp=sharing)
>
> : (From left to right) the flow map between the two frames, the previous frame, and the current frame of the task "Walker, walk," of the DM Control Suite.
>
> **Additional evaluation metric**
>
> In all experiments, we conducted the evaluation on Atari 100K by measuring the performance with 10 or more random seeds, following previous studies including SPR [1] and CURL [2]. Recently, a paper [3] published in NeurIPS 2021 analyzes the problems related to statistical uncertainty in the existing evaluation method of Atari 100K, as one reader recommended. Accordingly, we will add the results of a more complete evaluation using [3] to the appendix of the revised paper. For the evaluation result of using "Probability of Improvement" proposed in [3], please refer to the reply to the author of [3] above.
>
> We deeply appreciate your advice.
>
> [1] Data-efficient reinforcement learning with self-predictive representations. In International Conference on Learning Representations (ICLR), 2021.
>
> [2] CURL: contrastive unsupervised representations for reinforcement learning. In International Conference on Machine Learning (ICML), pp. 5639– 5650, 2020b.
>
> [3] Deep reinforcement learning at the edge of the statistical precipice. In NeurIPS, 2021.

---

### Official Review · Reviewer_Wtop · 2021-11-02

**Correctness:** 4
**Technical Novelty And Significance:** 2
**Empirical Novelty And Significance:** 2
**Recommendation:** 5
**Confidence:** 2

**Main Review:**

The paper is very well-written and referenced, and the figures are quite clear (especially Fig 2 which gives a great overview of the method). I find the idea of leveraging self-supervised flow models quite interesting as an alternative to brute-force next-step prediction.

My main concern, however, is that the results the authors chose to put forward are not as good as I'd have liked, which makes me doubt the overall significance of the method:
- in Atari, the proposed S³R performs significantly better than SPR in 12/26 tasks, and underperforms significantly in 7/26 tasks;
- in Control suite at 100K steps, S³R significantly outperforms DRQ in just one task out of 6, and underperforms significantly in 2, and both are more-or-less tied at 500K steps;
Since S³R seems significantly more complex in terms of implementation and computational cost, it seems unlikely that a practitioner in the field would bother compared implementing it rather than using a much simpler variant.

Some remarks/questions:
- I'm not aware of a definition of "structured representation", but I'm not sure I'm convinced by the claim that S³R's repr is more structured  because its features translate to flow maps. I also don't think it's necessary for the paper to make this claim.
- Introduction: "Learning visual features from raw pixels only using a reward function leads to limited performance and low sample efficiency."  I would tame down/remove that claim. I don't think there's any consensus on that, and work such as  https://arxiv.org/pdf/2104.06294.pdf are able to improve performance and sample efficiency without relying on an explicit visual features representation learning scheme
- I was surprised by the results of Table 3 (that F+W > P, and that F+W+P > F+W), as I would have expected next-step latent prediction to be a somewhat easy task. Do the authors have any further insight on this, eg on how (dis)similar Z^w and Z^p are, maybe in terms of an auxiliary stop-gradiented Sim operator?
- Following up on that, could using both Z^w and Z^q as input to the RL head bring an improvement?
- Notations: the paper (eg fig 2 and the paragraph at the top of p5) seems to imply that a_k is computed from I_k to I_{k+M}, should this be I_{k-M} to I_k instead?

**Summary Of The Paper:**

This paper focuses on self-supervised visual representation learning for RL applications. As opposed to previous literature that focused on learning global representations for the current observation, the authors propose learning "structured" representations based on flow maps that encode local structure. To do so, they combine:
- a self-supervised flow model (from previous literature), trained using an image reconstruction loss;
- a classic action-conditioned 1-step transition model using cosine similarity in latent space
- a flow based warping (also from previous literature) to provide a second next-step latent prediction task.
They evaluate their method on various standard RL tasks in two domains (Atari, DM Control Suite) and show improvements over SOTA in ~half of the tasks.

**Summary Of The Review:**

The paper combines existing self-supervised architectures (self-supervised flow estimation and InfoNCE-like similarity loss) to learn visual features in a rather novel and intriguing combination, but the overall increment in RL performance does not seem to justify the added complexity compared to much simpler counterparts.

---

> ### Author Response · Authors · 2021-11-21
> **References**
>
> [1] Alexey Dosovitskiy, Philipp Fischer, Eddy Ilg, Philip Häusser, Caner Hazirbas, Vladimir Golkov, Patrick van der Smagt, Daniel Cremers, and Thomas Brox. Flownet: Learning optical flow with convolutional networks. In IEEE International Conference on Computer Vision (ICCV), pp. 2758–2766, 2015.
>
> [2] Michael Laskin, Aravind Srinivas, and Pieter Abbeel. CURL: contrastive unsupervised representations for reinforcement learning. In International Conference on Machine Learning (ICML), pp. 5639– 5650, 2020b.
>
> [3] Max Schwarzer, Ankesh Anand, Rishab Goel, R Devon Hjelm, Aaron Courville, and Philip Bachman. Data-efficient reinforcement learning with self-predictive representations. In International Conference on Learning Representations (ICLR), 2021.
>
> [4] Agarwal, R., Schwarzer, M., Castro, P.S., Courville, A. and Bellemare, M.G., Deep reinforcement learning at the edge of the statistical precipice. In NeurIPS, 2021.
>
> [5] Michael Laskin, Kimin Lee, Adam Stooke, Lerrel Pinto, Pieter Abbeel, and Aravind Srinivas. Reinforcement learning with augmented data. In Advances in Neural Information Processing Systems (NeurIPS), 2020a.
>
> [6] Ilya Kostrikov, Denis Yarats, and Rob Fergus. Image augmentation is all you need: Regularizing deep reinforcement learning from pixels. arXiv preprint arXiv:2004.13649, 2020.
>
> [7] Schrittwieser, Julian, et al. "Online and offline reinforcement learning by planning with a learned model." arXiv preprint arXiv:2104.06294 (2021).

---

> ### Author Response · Authors · 2021-11-21
> **Responses to remaining concerns**
>
> **Q. I'm not aware of a definition of "structured representation", but I'm not sure I'm convinced by the claim that S³R's repr is more structured because its features translate to flow maps. I also don't think it's necessary for the paper to make this claim.**
>
> : We use the "structured representations" with the intention to indicate 3D volume which consists of 2D spatial coordinate and 1D feature dimension. However, we understand your concerns about using the terminology "structured representation". If you approve, we would like to change the terminology to a more agreeable expression such as "spatial representations" in the revised paper.
>
> **Q. "Learning visual features from raw pixels only using a reward function leads to limited performance and low sample efficiency." I would tame down/remove that claim. I don't think there's any consensus on that.**
>
> : We appreciate your suggestion. Based on the studies [5, 6, 7], we recognize that this claim is difficult to obtain consent. We will revise this sentence as follows: "Without designing an effective algorithm that uses model-based policy and value improvement operators [7], or without attempting to use additional image augmentation [5, 6], learning visual features from raw pixels only using a reward function might fail to learn good features in terms of the performance and sample efficiency."
>
> **Q. I was surprised by the results of Table 3 (that F+W > P, and that F+W+P > F+W), as I would have expected next-step latent prediction to be a somewhat easy task. Do the authors have any further insight on this, eg on how (dis)similar $Z^w$ and $Z^p$ are, maybe in terms of an auxiliary stop-gradiented Sim operator?**
>
> : $Z^w$ is created by a warping operation using a flow map that is predicted using $Z_k$ and the actual $Z_{k+1}$, while $Z^p$ is predicted by using $Z_k$ and corresponding action $a_k$. Thus, the flow based warping, which can use the actual data at ${(k+1)}^{th}$ instance, may be a better constraint than the transition model based prediction. Also, the representation learning in the flow based warping of 'F+W' makes the network learn representations more effectively.
>
> In addition, in the case of $Z^p$, M frames are stacked and entered into the transition model to predict all the next frames at once, but in the case of $Z^w$, each frame is encoded and warped using a flow vector individually. As such, stacking before conversion in $Z^p$ causes some information loss, and we analyze this may also affect the performance gap slightly.
>
> **Q. Could using both $Z^w$ and $Z^q$ as input to the RL head bring an improvement?**
>
> : We appreciate your suggestion. However, since only the current frame's representation must be entered into the RL head, the  generated $Z^w_{k+1}$ at the next frame (k+1) is unlikely to be usable. Also, note that the flow based warping is not used at an inference, and thus $Z^q_k$ is a sole representation for the RL MLP head.
>
> **Q. The paper (eg fig 2 and the paragraph at the top of p5) seems to imply that $a_k$ is computed from $I_k$ to $I_{k+M}$, should this be $I_{k-M}$ to $I_k$ instead?**
>
> : We expect that what you marked as "$a_k$" is actually "$o_k$" which means raw observation. $o_k$ is not computed through any operation, but it simply expresses the stack of multiple images, same as the existing methods in the field. As you mentioned, it is fine to use from $I_{k-M}$ to $I_k$. In this case, the flow maps should be computed with the previous frames.
>
> **Additional evaluation metric**
>
> In all experiments, we conducted the evaluation on Atari 100K by measuring the performance with 10 or more random seeds, following previous studies including SPR [3] and CURL [2]. Recently, a paper [4] published in NeurIPS 2021 analyzes the problems related to statistical uncertainty in the existing evaluation method of Atari 100K, as one reader recommended. Accordingly, we will add the results of a more complete evaluation using [4] to the appendix of the revised paper. For the evaluation result of using "Probability of Improvement" proposed in [4], please refer to the reply to the author of [4] above.
>
> Finally, we would like to express our deepest gratitude for sharing the various implications of our paper and for devoting your valuable time.

---

> ### Author Response · Authors · 2021-11-21
> **Response related to Computational cost and Performance (Part 2)**
>
> **DM Control Suite**: We are very aware of your concerns. First, the performance at 100K steps is usually based on when most methods do not converge. The performance of S³R recorded the highest mean in 2 tasks out of 6 tasks, and RAD [5] and DrQ [6] also recorded the highest mean in 2 tasks out of 6 tasks. It can be interpreted as RAD [5], DrQ [6] and S³R are the three methods with the highest convergence speed.
>
> In general, the performance at 500K steps after most methods converge is widely adopted in this field. Our method shows the highest performance in 4 out of 6 tasks compared to state-of-the-arts. Considering that this performance is the average value obtained by running 10 random seeds, it is a credible assessment. Also, when compared to the performance improvement rate of other methods, the performance increase of S³R is by no means small.
>
> However, as you pointed out, whether this performance gain is meaningful versus the increase in complexity may depend on the task being used. As mentioned above, our method has advantages over simple methods for tasks with complex structural features, and thus we believe the performance improvement by S³R is well worth.
>
> Our paper was in a great need of such additional analysis of complexity and performance, respectively. We will attach the above explanation as an independent section in the Appendix of the revised paper.
>
> Furthermore, in addition to the performance improvement, we would like to point out that our work is the first to unify computer vision tasks and reinforcement learning (RL) models and train them end-to-end. We hope the research in this direction could lead to new breakthroughs in both communities and our pioneering work can be a corner stone for opening this new opportunity. To be specific, the computer vision tasks aiming to effectively encode visual features can play an important role for visual RL algorithms using raw visual data as inputs.

---

> ### Author Response · Authors · 2021-11-21
> **Response related to Computational cost and Performance (Part 1)**
>
> We highly appreciate your valuable feedback and constructive criticism. Below, we address your points individually.
>
> **Q. The results the authors chose to put forward are not as good as I'd have liked, which makes me doubt the overall significance of the method. (Atari, DMControl Suite)**
>
> : We also deeply agree with what you pointed out. First, we quantitatively show the computational cost and performance increase of S³R in details.
>
> - Computational cost
>
> This increase in the computational cost for training is unavoidable because we additionally leverage the flow prediction and warping networks used in the vision task, but we found that the additional computational cost for training is not so significant. For training on DM Control Suite up to 500K on the same GPU environment, the proposed method takes about 16 hours, whereas the state-of-the-art methods CURL [2] and SPR [3] take about 10 hours and 13 hours respectively. Note that the original SPR paper did not provide the code implemented for DM Control Suite, so we conducted the experiments by slightly modifying the original SPR code. Additionally, the flow prediction and warping networks are used only during training, and the inference process is implemented in the same manner as other methods. Therefore, the inference time of our method is exactly the same as that of the state-of-the-arts methods (CURL [2], SPR [3], DrQ [6]) as long as the same encoder for query images is used. We will include the complexity comparison in the revised paper.
>
> - Performance increase of S³R
>
> **26 Atari Games**: Based on the evaluation method used in the existing Atari data, CURL [2] recorded the highest mean in 7 games out of 26, and SPR [3] recorded the highest mean in 11 games out of 26. Our S³R has the highest mean in 13 games out of 26. It can be interpreted that the performance increase of the S³R is not small by considering the quantitative aspects of these games, as also pointed out by other reviewers. Also, among the 13 games in which S³R has an edge, in particular, in 8 games (Alien, Assault, Gopher, Jamesbond, Krull, Kung Fu Master, Ms Pacman, and Seaquest), our method records a remarkably higher performance compared to other methods. Even the performance of certain games is high enough to match that of humans. This is because our method of capturing the local spatial structure is able to derive an effective representation from the images of the specific Atari games with various movements.
>
> However, as your pointed out, S³R may not be effective for some games. In particular, S³R did not perform well in the task "Pong". The biggest reason for this is that there are too few *discriminative* spatial structures available in the game images. Therefore, we can be sure that our representation learning method, which effectively captures the spatial structure, will work particularly well for data with much more complex structural features. In other words, while most of the simple methods suffer from training with data with complex structural features, our method can be a good substitute for addressing this.
>
> A method with relatively low-complexity may be preferred depending on the situation. However, as mentioned above, it was shown from the Atari benchmark that it is a much better choice to use S³R in the tasks that need to capture complex structural features.

---

### Official Review · Reviewer_pMqF · 2021-11-02

**Correctness:** 4
**Technical Novelty And Significance:** 2
**Empirical Novelty And Significance:** 3
**Recommendation:** 6
**Confidence:** 3

**Main Review:**

Strengths:

1. Quantitative results are promising. The proposed approach clearly shows promise over prior approaches.

2. Evaluation is thorough. The results hold across different environments and even different agent setups (Rainbow and SAC)

3. The paper is well-motivated on a conceptual and mathematical level.

Weaknesses:

1. The technical novelty is limited. The approach is very similar to SPR with the addition of latent variable warping.

2. It's difficult to conceive how this method could apply to different input modalities (i.e. audio, text, joint positions). The flow-based approach is very much tied to image inputs.

3. While there is improvement in quantitative performance over baselines, the margin of improvement is fairly small.

**Summary Of The Paper:**

In this paper, the authors focus on the problem of representation learning for deep reinforcement learning. To this end, they propose an approach to learning structured representations via establishing flows between latent volumes. Similar to SPR, they predict future representations with a transition model conditioned on actions. They evaluate their approach in two domains: Atari and DeepMind Control Suite and show improved performance over the state of the art.

**Summary Of The Review:**

The authors propose an approach to self-supervised learning for reinforcement learning which is very similar to SPR. The key contribution is the addition of a flow-based approach to predicting future latent vectors. Because of this, the generality of the approach is limited to image inputs. However, the results are promising and hold across different domains and agent setups.

---

> ### Author Response · Authors · 2021-11-21
> **Response (Part 2)**
>
> **26 Atari Games**: Based on the evaluation method used in the existing Atari data, CURL [4] recorded the highest mean in 7 games out of 26, and SPR [1] recorded the highest mean in 11 games out of 26. Our S³R has the highest mean in 13 games out of 26. It can be interpreted that the performance increase of the S³R is not small by considering the quantitative aspects of these games, as also pointed out by other reviewers. Also, among the 13 games in which S³R has an edge, in particular, in 8 games (Alien, Assault, Gopher, Jamesbond, Krull, Kung Fu Master, Ms Pacman, and Seaquest), our method records a remarkably higher performance compared to other methods. Even the performance of certain games is high enough to match that of humans. This is because our method of capturing the local spatial structure is able to derive an effective representation from the images of the specific Atari games with various movements.
>
> However, S³R may not be effective for some games. In particular, S³R did not perform well in the task "Pong". The biggest reason for this is that there are too few discriminative spatial structures available in the game images. Therefore, we can be sure that our representation learning method, which effectively captures the spatial structure, will work particularly well for data with much more complex structural features. In other words, while most of the simple methods suffer from training with data with complex structural features, our method can be a good substitute for addressing this.
>
> Our paper was in great need of such additional analysis of complexity and performance, respectively. We will attach the above explanation as an independent section in the Appendix of the revised version of the paper, and our result will become much more complete. We deeply appreciate it.
>
> **Additional evaluation metric**
>
> In all experiments, we conducted the evaluation on Atari 100K by measuring the performance with 10 or more random seeds, following previous studies including SPR [1] and CURL [4]. Recently, a paper [5] published in NeurIPS 2021 analyzes the problems related to statistical uncertainty in the existing evaluation method of Atari 100K, as one reader recommended. Accordingly, we will add the results of a more complete evaluation using [5] to the appendix of the revised paper. For the evaluation result of using "Probability of Improvement" proposed in [5], please refer to the reply to the author of [5] above.
>
> **References**
>
> [1] Max Schwarzer, Ankesh Anand, Rishab Goel, R Devon Hjelm, Aaron Courville, and Philip Bachman. Data-efficient reinforcement learning with self-predictive representations. In International Conference on Learning Representations (ICLR), 2021.
>
> [2] Michael Laskin, Kimin Lee, Adam Stooke, Lerrel Pinto, Pieter Abbeel, and Aravind Srinivas. Reinforcement learning with augmented data. In Advances in Neural Information Processing Systems (NeurIPS), 2020a.
>
> [3] Ilya Kostrikov, Denis Yarats, and Rob Fergus. Image augmentation is all you need: Regularizing deep reinforcement learning from pixels. arXiv preprint arXiv:2004.13649, 2020.
>
> [4] Michael Laskin, Aravind Srinivas, and Pieter Abbeel. CURL: contrastive unsupervised representations for reinforcement learning. In International Conference on Machine Learning (ICML), pp. 5639– 5650, 2020b.
>
> [5] Agarwal, R., Schwarzer, M., Castro, P.S., Courville, A. and Bellemare, M.G., Deep reinforcement learning at the edge of the statistical precipice. In NeurIPS, 2021.

---

> ### Author Response · Authors · 2021-11-21
> **Response (Part 1)**
>
> We highly appreciate your positive review and constructive criticism.
>
> **Q. The technical novelty is limited. The approach is very similar to SPR with the addition of latent variable warping.**
>
> : SPR [1] proposed to adopt the contrastive loss [4] for imposing the similarity between the predicted future representation from the action-conditioned transition model and the future representation of target encoder. It considers only the global-level similarity between two representations. In contrast, we first attempted to capture the change in terms of spatial structures, i.e., flow maps among multiple frames, for encoding local spatial structures, and also measure the global-level similarity between the warped representation and the future representation of target encoder. As reported in Table 3, the scores of using flow-based local similarity and warping based global similarity, "F+W", is already better than the scores of using the action-conditioned transition model, "P" which can be seen as SPR results. We further attempt to leverage both the warping based representation model and the action-conditioned transition model in a unified fashion, which further boosts the overall performance ("F+W+P" > "F+W" > "P"). Incidentally, while SPR verified the performance only for Atari, S³R evaluated the performance for various task domains such as Atari100K and DMControl Suite 100K/500K.
>
> Furthermore, in addition to the technical novelty, we would like to point out that our work is the first to unify computer vision tasks and reinforcement learning (RL) models and train them end-to-end. We hope the research in this direction could lead to new breakthroughs in both communities and our pioneering work can be a corner stone for opening this new opportunity. To be specific, the computer vision tasks aiming to effectively encode visual features can play an important role for visual RL algorithms using raw visual data as inputs.
>
> **Q. It's difficult to conceive how this method could apply to different input modalities (i.e. audio, text, joint positions). The flow-based approach is very much tied to image inputs.**
>
> : You are right. Indeed, this study is based on flow estimation and is difficult to apply to other input modalities such as audio, text, and joint positions. The main objective of this work is to introduce a new methodology for unifying computer vision tasks and RL algorithms. Though this work is much tied to image inputs, we believe its importance is still significant considering visual RL is an active research area. Additionally, inspired by this work, we hope that the self-supervised audio/text learning algorithms could be employed in the RL algorithms. We reserve this as future work.
>
> **Q. While there is improvement in quantitative performance over baselines, the margin of improvement is fairly small.**
>
> We also deeply agree with what you pointed out. Below, we address your points on the two benchmark in terms of the performance improvement.
>
> **DM Control Suite**: We are very aware of your concerns. First, the performance at 100K steps is usually based on when most methods do not converge. The performance of S³R recorded the highest mean in 2 tasks out of 6 tasks, and RAD [2] and DrQ [3] also recorded the highest mean in 2 tasks out of 6 tasks. It can be interpreted as RAD [2], DrQ [3] and S³R are the three methods with the highest convergence speed.
>
> In general, the performance at 500K steps after most methods converge is widely adopted in this field. Our method shows the highest performance in 4 out of 6 tasks compared to state-of-the-arts. Considering that this performance is the average value obtained by running 10 random seeds, it is a credible assessment. Also, when compared to the performance improvement rate of other methods, the performance increase of S³R is by no means small.

---

### Official Review · Reviewer_aHc1 · 2021-11-03

**Correctness:** 3
**Technical Novelty And Significance:** 3
**Empirical Novelty And Significance:** 3
**Recommendation:** 5
**Confidence:** 4

**Main Review:**

strengths:
* the method does outperform competitive baselines on challenging atari games
* the paper is well written and the method seems good.
* results are shown with a fixed set of loss weights which shows robustness.

weaknesses:
* the method is very complex and hard to reproduce. Even if the code is made available, building on top of this method will not be trivial.
* the ablation experiments are done on domains that favour good flow representations and lack baselines (dm control)
* the crux of the state-representation problem is that it should make credit assignment easier but the hard exploration games are either missing from the experiments or improvements seem marginal.

remarks:
* i think structured representation is a misnomer here. This is because the image lattice is the weakest form of structure you could have and because structured representations in the literature refer to objects and parts. So I recommend changing the title to reflect that.
* i can't seem to find the L_rl loss in your text. Is it SAC ? I guess it's just above the loss functions where you hint that both DQN and SAC where used somehow
* The method seems to struggle with pong ? Is this because the RL loss in this setting and these hyper parameters is washed away by the auxiliary loss gradients which have a hard time converging ?


**Summary Of The Paper:**

This paper proposes a representation learning method that leverages unsupervised signals like flow and forward models and the constraints between them and apply it to the state representation problem of RL.

The method is adding a number of auxiliary rewards to the torso which flow and a latent transition model that are constrained to agree with a contrastive loss. The flow has an architecture inspired by FlowNet and trained with warping reconstruction and a spatial smoothness term. A latent transition model is trained contrastively. Then constraints are added to make the latent of the next step agree with both of these predictions.

The method is run on 13 Atari tasks and compared there with competitive baselines and it largely outperforms them. An ablation without baselines is performed a few dm control suite tasks

**Summary Of The Review:**

The method seems reasonable but the approach is very complicated and there are some lacking experiments that would make the claims more compelling. If the community is to be convinced to embrace such a complicated approach experiments need to be more compelling i should think.

---

> ### Author Response · Authors · 2021-11-21
> **References**
>
> [1] Max Schwarzer, Ankesh Anand, Rishab Goel, R Devon Hjelm, Aaron Courville, and Philip Bachman. Data-efficient reinforcement learning with self-predictive representations. In International Conference on Learning Representations (ICLR), 2021.
>
> [2] Michael Laskin, Kimin Lee, Adam Stooke, Lerrel Pinto, Pieter Abbeel, and Aravind Srinivas. Reinforcement learning with augmented data. In Advances in Neural Information Processing Systems (NeurIPS), 2020a.
>
> [3] Ilya Kostrikov, Denis Yarats, and Rob Fergus. Image augmentation is all you need: Regularizing deep reinforcement learning from pixels. arXiv preprint arXiv:2004.13649, 2020.
>
> [4] Michael Laskin, Aravind Srinivas, and Pieter Abbeel. CURL: contrastive unsupervised representations for reinforcement learning. In International Conference on Machine Learning (ICML), pp. 5639– 5650, 2020b.
>
> [5] Agarwal, R., Schwarzer, M., Castro, P.S., Courville, A. and Bellemare, M.G., Deep reinforcement learning at the edge of the statistical precipice. In NeurIPS, 2021.

---

> ### Author Response · Authors · 2021-11-21
> **Response (Part 2)**
>
> **26 Atari Games**: Based on the evaluation method used in the existing Atari data, CURL [4] recorded the highest mean in 7 games out of 26, and SPR [1] recorded the highest mean in 11 games out of 26. Our S³R has the highest mean in 13 games out of 26. It can be interpreted that the performance increase of the S³R is not small by considering the quantitative aspects of these games, as also pointed out by other reviewers. Also, among the 13 games in which S³R has an edge, in particular, in 8 games (Alien, Assault, Gopher, Jamesbond, Krull, Kung Fu Master, Ms Pacman, and Seaquest), our method records a remarkably higher performance compared to other methods. Even the performance of certain games is high enough to match that of humans. This is because our method of capturing the local spatial structure is able to derive an effective representation from the images of the specific Atari games with various movements.
>
> However, S³R may not be effective for some games. In particular, S³R did not perform well in the task "Pong". The biggest reason for this is that there are too few discriminative spatial structures available in the game images. Therefore, we can be sure that our representation learning method, which effectively captures the spatial structure, will work particularly well for data with much more complex structural features. In other words, while most of the simple methods suffer from training with data with complex structural features, our method can be a good substitute for addressing this.
>
> Our paper was in great need of such additional analysis of complexity and performance, respectively. We will attach the above explanation as an independent section in the Appendix of the revised version of the paper, and our result will become much more complete. We deeply appreciate it.
>
> **Q. I think structured representation is a misnomer here. This is because the image lattice is the weakest form of structure you could have and because structured representations in the literature refer to objects and parts. So I recommend changing the title to reflect that.**
>
> : We use the "structured representations" with the intention to indicate 3D volume which consists of 2D spatial coordinate and 1D feature dimension. However, we understand your concerns about using the terminology "structured representation". If you approve, we would like to change the terminology to a more agreeable expression such as "spatial representations" in the revised paper.
>
> **Q. i can't seem to find the L_rl loss in your text. Is it SAC ? I guess it's just above the loss functions where you hint that both DQN and SAC where used somehow**
>
> : In the first paragraph of the Section "Method", we specified the RL loss as follows.
>
> “Following the state-of-the-arts RL approaches using the self-supervised learning, we adopt the Soft Actor Critic (SAC) method for continuous control task in DeepMind Control Suite benchmark, and Rainbow DQN for discrete control task in Atari Games.”
>
> We apologize for the confusion in understanding this. We will put the above explanation in several places so that readers can better understand the proposed method.
>
> **Q. The method seems to struggle with pong ? Is this because the RL loss in this setting and these hyper parameters is washed away by the auxiliary loss gradients which have a hard time converging ?**
>
> As mentioned above, S³R did not perform well in the task "Pong". The biggest reason for this is that there are too few *discriminative* spatial structures available in the game images. Therefore, we can be sure that our representation learning method, which effectively captures the spatial structure, will work particularly well for data with much more complex structural features.
>
> **Additional evaluation metric**
>
> In all experiments, we conducted the evaluation on Atari 100K by measuring the performance with 10 or more random seeds, following previous studies including SPR [1] and CURL [4]. Recently, a paper [5] published in NeurIPS 2021 analyzes the problems related to statistical uncertainty in the existing evaluation method of Atari 100K, as one reader recommended. Accordingly, we will add the results of a more complete evaluation using [5] to the appendix of the revised paper. For the evaluation result of using `Probability of Improvement’ proposed in [5], please refer to the reply to the author of [5] above.

---

> ### Author Response · Authors · 2021-11-21
> **Response (Part 1)**
>
> We highly appreciate your valuable feedback and constructive criticism. Below, we address your points individually.
>
> **Q. The method is very complex and hard to reproduce. Even if the code is made available, building on top of this method will not be trivial.**
>
> : We deeply understand your concerns, but we would like to say that our method can be interpreted more easily than you think. In Figure 1 of our original manuscript, you can find the abstract version of the method. We predict the representation corresponding to the next time instance using flow based warping and the transition model, and then give them both global-level and pixel-level similarity constraints with the targeted representation. Simply put, that’s all. The distinctiveness of our method comes from the spatial representation that we use for encoding local spatial structures from the consecutive frames, and the method of using this representation itself is not complicated.
>
> Also, the proposed method can be incorporated into any kind of representation learning methods due to its off-the-shelf nature. In the environment using consecutive frames, if an auxiliary cost function to predict the next frame is used along with the self-supervised flow estimation model and the state transition model, the performance improvement of various methods can be derived.
>
> **Q. The ablation experiments are done on domains that favour good flow representations and lack baselines (dm control)**
>
> : We are not sure whether we fully understood your concerns, but several results of DM Control Suite benchmarks, including the baseline, are provided in Table 2 and 3, following the evaluation metric used in the existing methods. Also, from Table 2, it cannot be seen that all 6 tasks favour good flow representations, and therefore we think that our ablation study was conducted in a fairly impartial environment.
>
> **Q. The crux of the state-representation problem is that it should make credit assignment easier but the hard exploration games are either missing from the experiments or improvements seem marginal.**
>
> : In this paper, for a fair comparison, we have verified the proposed method with the tasks commonly adopted by the existing methods based on the representation learning in the field of RL. In the future works, we are also willing to verify our method for harder exploration games, as you mentioned.
>
> Regarding the performance improvement by our method, we address your points on the two benchmark in terms of the performance improvement.
>
> **DM Control Suite**: First, the performance at 100K steps is usually based on when most methods do not converge. The performance of S³R recorded the highest mean in 2 tasks out of 6 tasks, and RAD [2] and DrQ [3] also recorded the highest mean in 2 tasks out of 6 tasks. It can be interpreted as RAD [2], DrQ [3] and S³R are the three methods with the highest convergence speed.
>
> In general, the performance at 500K steps after most methods converge is widely adopted in this field. Our method shows the highest performance in 4 out of 6 tasks compared to state-of-the-arts. Considering that this performance is the average value obtained by running 10 random seeds, it is a credible assessment. Also, when compared to the performance improvement rate of other methods, the performance increase of S³R is by no means small.

---

> > ### Comment · Reviewer_aHc1 · 2021-12-03
> > **response**
> >
> > Thank you for the response.
> >
> > As I said in the original review the method itself makes sense, that was never the question, I am highlighting more that producing a good flow model that performs well is not trivial in practice. There are reasons why supervised synthetic flow is still outperforming unsupervised flow in the literature, why a new SOTA in optical flow gets best paper at ECCV etc. The models and losses are very fickle and architectures are hard to get right on non-trivial data. In some sense sprite data like Atari and dm control may be why an unsupervised method like the one proposed actually worked so well.
> >
> > Setting that aside though I find the method too complicated for the gains it offers especially compared to DrQ which is a really easy to implement technique and requires minimal knowledge to get to work. Nonetheless, I do acknowledge the novelty so I have to leave the AC with this tough decision. From my point of view I could easily support both acceptance and rejection.

---

### Public Comment · ~Rishabh_Agarwal2 · 2021-11-09
**Statistical uncertainty in reported results**

Hi authors,

The case study in [1] on Atari 100k shows that results on this benchmark has significant variance in results with overlapping standard deviations for various games and even changes in ranking comparisons across methods when using 100 seeds. See [this figure](https://pbs.twimg.com/media/E-IkSDdXMAUV1dx?format=jpg&name=large) for an example. However, the current results for Atari 100k ignore the statistical uncertainty in results and simply report the mean scores.

I'd recommend the authors to follow the reliable evaluation protocols suggested in [1] when using only a few seeds such as reporting score distributions and aggregate performance metrics like IQM with confidence intervals (CIs). Similarly, the number of games with best ranking can be replaced by the probability of being the best ranked -- the probability of improvement metric with CIs can be a simpler alternative to show the same thing. You can easily do so using the library at https://github.com/google-research/rliable or the [colab](https://bit.ly/statistical_precipice_colab).

Minor point: CURL achieves a lower score than reported as CURL's reported scores were based on maximum performance during training rather than end performance results. Similarly, DrQ(\epsilon) which uses standard values for e-greedy would be a more fair comparison. Please find the results for 100 runs for prior Atari 100k methods here: https://console.cloud.google.com/storage/browser/rl-benchmark-data/atari_100k.

[1] Agarwal, R., Schwarzer, M., Castro, P.S., Courville, A. and Bellemare, M.G., 2021. Deep reinforcement learning at the edge of the statistical precipice. In NeurIPS.

---

> ### Author Response · Authors · 2021-11-21
> **Response**
>
> We appreciate your interest in this paper, and thank you very much for providing useful information.
>
> For a fair comparison, we followed the evaluation method used in previous methods. However, your claim that there is statistical uncertainty in the existing evaluation method seems valid, and accordingly, we will add the evaluation results using the method proposed in [1] to our Appendix.
>
> Below are the evaluation results conducted with "Probability of Improvement" proposed in [1].
>
> [Figure](https://drive.google.com/file/d/1yzP-RtPEPC_OJUWsCzfDXD4aWH1_pJpX/view?usp=sharing)
>
> Quoting the description in [1] to help reviewers and other readers understand, this evaluation method estimates how likely an algorithm improves upon another algorithm. For instance, "P(SimPLe>X)" indicates the probability (written in the horizontal line) that "SimPLe" is better than another algorithm, called "X", which is listed in the vertical line (e.g., S³R, SPR,..., DER). Namely, the probability that SimPLe is better than SPR, "P(SimPLe>SPR)", has an average probability of about 0.3.
>
> In this context, in the six graphs above, the smaller the value of S³R, the higher the performance. Also, in the graph of S³R below, "P(S³R>X)", S³R will rank the highest when the remaining bars of algorithms are located to the right of the bar of S³R. From this analysis, it can be reconfirmed that S³R has the most superior performance compared to the state-of-the-art methods on Atari 100K as mentioned in the original manuscript. We add the results of a more complete evaluation using [1] to the Appendix of the revised paper.
>
> We appreciate you for conducting various analyzes to advance this field, and for giving us thoughtful advice.
>
> **Reference**
>
> [1] Agarwal, R., Schwarzer, M., Castro, P.S., Courville, A. and Bellemare, M.G., Deep reinforcement learning at the edge of the statistical precipice. In NeurIPS, 2021.

---

### Author Response · Authors · 2021-11-23
**Paper Revision**

Thanks to all the reviewers, we were able to revise the paper delicately. We have uploaded the revised version of the paper. The paper has been enriched with all the progressive comments, and if there are any further concerns, please let us know at any time.

The revisions to the paper reflecting the comments of reviewers are as follows:

- The terminology “structured representation” has changed to “spatial representation”. (**Reviewer Wtop**, **Reviewer aHc1**)
- We revised the sentence in the introduction as “Without designing an effective algorithm that uses model-based policy and value improvement operators (Schrittwieser et al., 2021), or without attempting to use additional image augmentation (Laskin et al., 2020a; Kostrikov et al., 2020), learning visual features from raw pixels only using a reward function might fail to learn good features in terms of the performance and sample efficiency.” (**Reviewer Wtop**)
- Further explanation of using the internal flow map was added in Method 3.2 and Appendix A.4. (**Reviewer iodY**)
- Flow map visualization results were added in Appendix A.4. (**Reviewer iodY**)
- Additional results using the evaluation metric proposed in Agarwal et al. (2021) were added in Appendix D. (**Rishabh Agarwal**)
- Further analysis on the computational cost and performance improvement was added in Appendix E. (**Reviewer Wtop**)
- Further analysis on the performance improvement was added in Appendix E. (**Reviewer pMqF**, **Reviewer aHc1**)

For other matters, please refer to our responses below. We thank the reviewers for their valuable comments.

---

### Decision · Program_Chairs · 2022-01-20

**Decision:**

Reject

**Comment:**

After carefully reading the reviews and the rebuttal, unfortunately I feel this work is not yet ready for acceptance.

I want to acknowledge the effort put in the rebuttal for this work and I think all the changes greatly increased the value of the work. However, I feel that the work could greatly benefit from running on a different domain where the gain is more considerable. My worry is that the complexity of the method compared to the relatively small improvement (at least as perceived from the current results) will reduce considerably the attention the work will receive from the community (unfairly so).
Or some of the analysis and ablation done (e.g. the flow visualization) which are now in the appendix could be brought in the main manuscript to be able to drive the message home. An understanding of the impact on the accuracy of the flow model on the overall performance (which as pointed out by reviewer aHc1 is a really hard task in a more natural context). Or maybe a 3D visually complex environment is exactly where this method will shine as flows are more complex and hence more informative.

Overall I think this is solid work, but I feel it does not manage to convince the reader of the significance of the proposed approach. And hence, if published in this form, I feel it will do a disservice to the work, as it will not receive the attention it merits from the community.